# S2GO: Streaming Sparse Gaussian Occupancy

**Jinhyung Park**[1,2][†]  **Chensheng Peng**[1,3][†]  **Yihan Hu**[1]  **Wenzhao Zheng**[3]
**Kris Kitani**[2]  **Wei Zhan**[1,3][‡]
[1]Applied Intuition  [2]Carnegie Mellon University  [3]University of California, Berkeley

## Abstract

Despite the efficiency and performance of sparse query-based representations for detection, state-of-the-art 3D occupancy estimation methods still rely on voxel-based or dense Gaussian-based 3D representations. However, dense representations are slow, and they lack flexibility in capturing the temporal dynamics of driving scenes. Distinct from prior work, we instead summarize the scene into a compact set of 3D queries which are propagated through time in an online, streaming fashion. These queries are then decoded into semantic Gaussians at each timestep. We couple our framework with a denoising rendering objective to guide the queries and their constituent Gaussians in effectively capturing scene geometry. Due to its efficient, query-based representation, S2GO achieves state-of-the-art performance on the nuScenes and KITTI occupancy benchmarks, outperforming prior art (e.g., GaussianWorld) by **2.7 IoU** with **4.5x faster** inference.

## 1 Introduction

Vision-centric autonomous systems provide a more cost-effective and scalable alternative to LiDAR-based solutions (Wayve, 2024; Mobileye, 2024; Tesla, 2022; Zhang et al., 2024), but they struggle with the absence of dense 3D geometry priors, an obstacle to achieving beyond Level 3 autonomy. To address this gap, 3D occupancy semantic estimation has emerged as a powerful complement to conventional sparse 3D perception tasks like bounding box detection (Philion & Fidler, 2020; Yin et al., 2021; Liu et al., 2022; Wang et al., 2023; Li et al., 2022b;a) or vectorized mapping (Li et al., 2021; Liao et al., 2022a; 2024; Chen et al., 2024; Yuan et al., 2024), because it captures a richer and more comprehensive view of unknown and arbitrarily shaped objects, thereby improving safety.

Recent 3D occupancy methods often rely on regular grids (Cao & de Charette, 2022; Li et al., 2022b; Zhang et al., 2023; Huang et al., 2023; Liu et al., 2024) or dense Gaussians (Huang et al., 2024; Zuo et al., 2025b; Zheng et al., 2024b). Although these methods capture high-fidelity details, they are slow and inflexible when integrating long-term historical context, limiting both static infrastructure localization as well as dynamic actor modeling. Existing grid-based approaches reduce redundancy by warping or projecting features from previous frames (Li et al., 2022b; Huang & Huang, 2021; Li et al., 2023b; Liu et al., 2023; 2024), but suffer from unnecessary computation in unoccupied regions and artifacts introduced by dense grids. Meanwhile, recent Gaussian-based techniques (Huang et al., 2024; 2025; Zheng et al., 2024b; Zuo et al., 2025b) show promise by focusing computation on occupied regions. However, they rely on tens of thousands of Gaussians (25.6k ∼ 144k) and use local sparse convolutions because global modeling becomes computationally prohibitive.

To address the inefficiencies of voxel-based and dense Gaussian-based methods in streaming perception, we propose to use *sparse 3D queries* to summarize and propagate the *dense 3D world* over time. More specifically, our method (**S2GO**) maintains a queue of past sparse 3D queries, refines the current set of queries using both previous queries and current image observations, and then predicts 3D occupancy by decoding the current queries into a denser set of semantic Gaussians. This online framework enables efficient propagation and global feature interaction among a sparser set of 3D queries (∼1k) while retaining the high fidelity of Gaussian-based representations.

Query-based perception has proven effective for sparse object detection (Carion et al., 2020; Wang et al., 2020; 2023), but employing sparse queries for dense, high-fidelity occupancy estimation presents several challenges. **First**, object detectors typically employ hundreds of queries, which far

---

[†]Work done during internship at Applied Intuition
[‡]Correspondence: wei.zhan@applied.co

outnumber the target objects (approximately 30 per scene), allowing for direct Hungarian Matching. In contrast, 3D occupancy estimation must cover the entire scene, making the mapping from sparse queries to dense semantic Gaussians inherently ambiguous. **Second**, in voxel-aligned occupancy estimation, fixed voxels simply perform classification at their predetermined locations. By comparison, query-based approaches require that queries first move to regions of interest before classifying. This creates a chicken-and-egg problem: for instance, if a query lies between a car and the road, it is unclear whether it should shift toward the car or the road, as the correct target location depends on the query's intended class. **Third**, while dense Gaussian methods mitigate this ambiguity through extensive spatial coverage, increasing the sparsity of the representation for efficiency exacerbates the difficulty of aligning queries accurately with occupied regions.

To fully unlock the streaming potential of query-based occupancy estimation, we introduce a pretraining phase that first trains the network to capture 3D scene geometry. During pretraining, query locations are initialized at noised LiDAR points, and the network is trained to recover 3D geometry through a denoising objective. To capture fine-grained local shape, decoded Gaussians are rendered from the current and neighboring views and supervised accordingly. The network also predicts a velocity for each query to model dynamic objects. This pretraining addresses the aforementioned challenges of using sparse queries by 1) supervising queries and their decoded Gaussians to model local scene structure, 2) training queries to self-organize to evenly cover the scene, and 3) supervising queries explicitly to move from empty space to occupied regions. Then, during the following semantic occupancy estimation stage – when LiDAR is no longer used and queries are randomly initialized throughout the 3D scene – the network uses its pretrained knowledge to precisely reposition the queries and decode to Gaussians to capture dense 3D structure.

Our contributions are summarized as follows.

- We present **S2GO**, a streaming framework for semantic occupancy where persistent, sparse queries carry long-horizon context and are decoded into dense occupancy via Gaussians.
- To address the challenge of making *dense* estimations from a *sparse* representation, we introduce a novel geometry denoising pretraining phase. This enables sparse queries to traverse empty space and self-organize onto occupied regions, yielding dense 3D structure.
- We further address key limitations in the oft-used Gaussian formulation and voxel splatting algorithm and *halve training time* with better performance.
- We evaluate S2GO on nuScenes and KITTI and achieve state-of-the-art performance and inference speed. Notably, our lightweight model improves over prior art (e.g. GaussianWorld) with **5.9**× faster inference, achieving real-time inference on a single 4090 (26 FPS).

## 2 RELATED WORK

**3D Occupancy Estimation** is increasingly crucial for vision-centric systems due to limited geometric priors inherent in purely vision-based methods. This task provides dense, volumetric representations of the environment, significantly enhancing semantic understanding and improving safety in decision-making, effectively complementing LiDAR. Recent camera-based benchmarks (Wei et al., 2023; Tian et al., 2023; Liao et al., 2022b), featuring detailed annotations created through offboard techniques, have driven substantial progress in vision-based occupancy modeling research.

Building on these benchmarks, existing methods (Cao & de Charette, 2022; Li et al., 2023b; Huang et al., 2023; Zhang et al., 2023; Ye et al., 2024; Zheng et al., 2024a; Tong et al., 2023) typically employ dense BEV or voxel-based representations, but such structures hinder real-time processing efficiency and scalability. Sparse voxel approaches (Liu et al., 2024; Li et al., 2023a; Wang et al., 2024b) enhance efficiency but encounter challenges such as complex temporal modeling and increased overhead in temporal integration due to their grid-based nature. One line of work He et al. (2025); Kim et al. (2025) leverage separate processing of voxels and BEV grids to capture both detail and context, and ProtoOcc Oh et al. (2025) lifts 2D prototypes to 3D and optimizes via contrastive loss. Recently, StreamOcc Moon et al. (2025) leverage both efficient 3D voxels and explicit instance queries with bi-directional aggregation. While achieving strong performance, the voxel-query interaction remains computationally expensive.

Recently, Gaussian-based representations (Kerbl et al., 2023; Peng et al., 2024; Yang et al., 2024; Xu et al., 2024) have emerged due to their strong 3D and semantic representational capabilities. Methods such as Huang et al. (2024); Zheng et al. (2024b); Zuo et al. (2025b) exploit probabilistic

semantic Gaussians for 3D occupancy modeling but require large numbers of Gaussians, posing challenges for real-time performance and efficient temporal fusion. Also related is OSP (Shi et al., 2024), which represents the scene as a set of points. While flexible, sparse points cover a narrower region of the scene compared to Gaussians, and OSP requires grid-aligned point sampling to make voxel-aligned predictions. The recent work QuadricFormer (Zuo et al., 2025a) makes efficient use of quadrics, but it still requires up to 12.8k independent primitives for competitive performance.

**Query-based Representations.** Since DETR (Carion et al., 2020), query-based methods have rapidly advanced, demonstrating effectiveness in tasks like detection, mapping, and tracking. DETR3D (Luo et al., 2022) efficiently extends 2D queries into 3D for detection, while Stream-PETR (Wang et al., 2023) fuses temporal information in a streaming fashion. MapTR (Liao et al., 2022a; 2024) leverages structured Transformers for HD map generation, and MapTracker (Chen et al., 2024; Yuan et al., 2024) reframes the mapping task with object tracking. Sparse4D (Lin et al., 2023) integrates detection and tracking into a unified, end-to-end framework. However, object-centric query methods remain underutilized for dense reconstruction tasks like occupancy estimation. We bridge this gap by introducing Gaussian queries, establishing a streamlined, query-based framework for efficient 3D semantic occupancy estimation.

## 3 S2GO

### 3.1 PRELIMINARY: GAUSSIAN OCCUPANCY ESTIMATION

GaussianFormer (Huang et al., 2024) and follow-up work (Huang et al., 2025; Zheng et al., 2024b; Zuo et al., 2025b) propose to represent the driving scene as a set of 3D Gaussian primitives $\mathcal{G} = \{\mathbf{G}_i\}_{i=1}^P$, with each semantic Gaussian $\mathbf{G}_i$ specified by its position $\mathbf{x}_i \in \mathbb{R}^3$, rotation $\mathbf{r}_i \in \mathbb{R}^4$, scale $\mathbf{s}_i \in \mathbb{R}^3$, opacity $a_i \in \mathbb{R}$ and class distribution $\mathbf{c}_i \in \mathbb{R}^C$, where $C$ is the number of foreground classes. Given this set, GaussianFormer-2 (Huang et al., 2025) predicts the semantic occupancy of a voxel coordinate $\mathbf{x} \in \mathbb{R}^3$ by first predicting binary occupancy probability and then expressing the class distribution of occupied regions as a mixture of nearby Gaussians. More specifically, the occupancy probability $\alpha(\mathbf{x}) \in \mathbb{R}$ is modeled as the probability that $\mathbf{x}$ is occupied by at least one of $P$ nearby Gaussians:

$$\alpha(\mathbf{x}) = 1 - \prod_{i=1}^P \big(1 - \alpha(\mathbf{x}; \mathbf{G}_i)\big) \tag{1}$$

where $\alpha(\mathbf{x}; \mathbf{G}_i)$ is the probability that $x$ is occupied by $\mathbf{G}_i$:

$$\alpha(\mathbf{x}; \mathbf{G}) = \exp\big(-\frac{1}{2}(\mathbf{x} - \mathbf{m})^{\mathrm{T}} \mathbf{\Sigma}^{-1} (\mathbf{x} - \mathbf{m})\big) \tag{2}$$

$$\mathbf{\Sigma} = \mathbf{R}\mathbf{S}\mathbf{S}^T\mathbf{R}^T, \quad \mathbf{S} = \mathrm{diag}(\mathbf{s}), \quad \mathbf{R} = \mathrm{q2r}(\mathbf{r}) \tag{3}$$

Further, the foreground class distribution $\mathbf{e}(\mathbf{x}; \mathcal{G}) \in \mathbb{R}^C$ is expressed as a mixture of Gaussians weighted by opacity $a$:

$$\mathbf{e}(\mathbf{x}; \mathcal{G}) = \sum_{i=1}^P p(\mathbf{G}_i|\mathbf{x})\tilde{\mathbf{c}}_i = \frac{\sum_{i=1}^P p(\mathbf{x}|\mathbf{G}_i)a_i\tilde{\mathbf{c}}_i}{\sum_{j=1}^P p(\mathbf{x}|\mathbf{G}_j)a_j}, \tag{4}$$

$$p(\mathbf{x}|\mathbf{G}_i) = \frac{1}{(2\pi)^{\frac{3}{2}} |\mathbf{\Sigma}|^{\frac{1}{2}}} \exp\big(-\frac{1}{2}(\mathbf{x} - \mathbf{m})^{\mathrm{T}} \mathbf{\Sigma}^{-1} (\mathbf{x} - \mathbf{m})\big) \tag{5}$$

Finally, the joint semantic occupancy distribution over foreground classes and the empty background is written as $[\alpha(\mathbf{x}) \cdot \mathbf{e}(\mathbf{x}; \mathcal{G}); 1 - \alpha(\mathbf{x})] \in \mathbb{R}^{(C+1)}$. We refer the reader to prior work (Huang et al., 2024; 2025) for additional details.

### 3.2 ARCHITECTURE

Our framework shown in Figure 1 is inspired by streaming query-based object detection methods (Carion et al., 2020; Wang et al., 2020; 2023; Lin et al., 2022; Yuan et al., 2024). We keep a queue of past sparse 3D queries, update the current queries based on historical queries and current images, and predict a detailed set of 3D Gaussians. The temporal transformer design follows PETR (Liu et al., 2022) and StreamPETR (Wang et al., 2023).

More specifically, at each timestep $t$, we represent the scene with a set of sparse 3D queries $\mathcal{Q}_t = \{\mathbf{q}_t^i\}_{i=1}^K$ with associated 3D locations $\{\mathbf{p}_t^i\}_{i=1}^K \subset \mathbb{R}^3$, where $K$ is the number of queries. These queries are refined using a queue of past queries $\bar{\mathcal{Q}}_t$ and the 2D features $\mathcal{F}_t = \mathrm{CNN}(I_t)$ from the RGB images of that timestep $I_t \in \mathbb{R}^{N \times H \times W \times 3}$, where $N$ is the number of cameras.

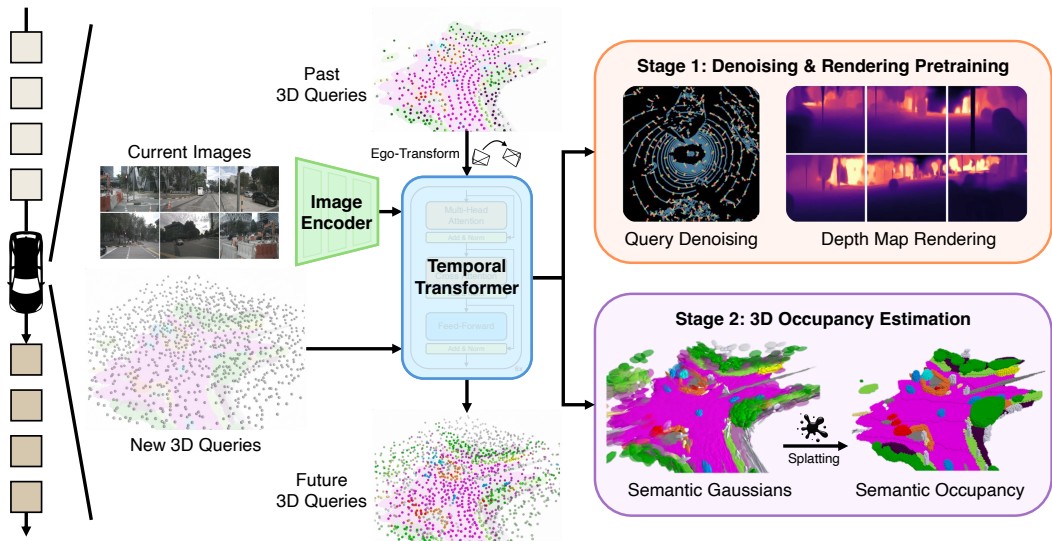

Figure 1: **Overall framework of S2GO for streaming perception.** At each timestep, our method refines new 3D queries using current image observations and a queue of past queries. These queries are decoded into a set of fine-grained Gaussians, and a portion of the queries are propagated to future timesteps in a streaming fashion. In Stage 1, this query refinement and Gaussian estimation pipeline is pretrained to effectively model the 3D scene using query denoising and rendering pretraining. In Stage 2, the predicted Gaussians are splatted to voxels for training 3D occupancy estimation.

Each query predicts a position offset $\mathbf{o}^i$, opacity $a^i$, and velocity $\mathbf{v}^i$, alongside attributes for a set of finer Gaussians. Relaxing the timestep $t$ subscript on Gaussians for clarity, the derived Gaussians are written as:

$$\mathcal{G}_t = \{\{(\mathbf{p}^i + \mathbf{o}^i + \mathbf{o}^i_j, \mathbf{v}^i, \mathbf{r}^i_j, \mathbf{s}^i_j, a^i \cdot a^i_j)\}_{j=1}^J\}_{i=1}^K \tag{6}$$

where $J$ is the number of Gaussians per query. Each Gaussian has a 3D position $\mathbf{p}^i + \mathbf{o}^i + \mathbf{o}^i_j$ combining the query position, query offset, and its own offset $\mathbf{o}^i_j$, a velocity $\mathbf{v}^i$ inherited from its parent query, a rotation $\mathbf{r}^i_j$, a scale $\mathbf{s}^i_j$, and an opacity $a^i \cdot a^i_j$ where the query opacity modulates the Gaussian-specific opacity $a^i_j$. This hierarchical decomposition allows each query to anchor a spatial region while the finer Gaussians capture local structure within that region.

Our framework for efficiently extracting 3D Gaussians from image observations is consistent across both the denoising pretraining and occupancy estimation tasks. The primary distinction lies in the additional attributes predicted by each Gaussian: during pretraining, each Gaussian independently predicts its own color, whereas, during occupancy estimation, Gaussians derived from the same query collectively share a semantic class label. This shared semantic class ensures consistency among Gaussians originating from a single query.

## 3.3 STAGE 1: 3D GEOMETRY DENOISING

### 3.3.1 MOTIVATION

While S2GO can directly be trained for occupancy estimation, the resulting performance is suboptimal. The queries and their Gaussians are unable to move effectively to occupied locations to capture fine details – they instead coarsely model nearby regions as shown in Figure 2. This stems from the weak and ambiguous supervision that queries and Gaussians receive from occupancy labels.

This limitation arises from two interconnected factors: **First**, unlike in GaussianFormer where each Gaussian is refined individually, in our sparse query-based framework, each query moves $J$ Gaussians as a group before individual Gaussians locally branch out. As any perturbations to query location propagate to its constituent Gaussians, aligning the query precisely with scene geometry before predicting Gaussian offsets is critical. However, 3D occupancy estimation lacks a clear assignment between parts of the scene and individual queries – with multiple nearby scene elements, the lack of clear-cut supervision causes query refinements to be noisy. **Second**, this ambiguity is exacerbated by the inherent *locality* of the Gaussian-to-voxel splatting operation in Section 3.1. As

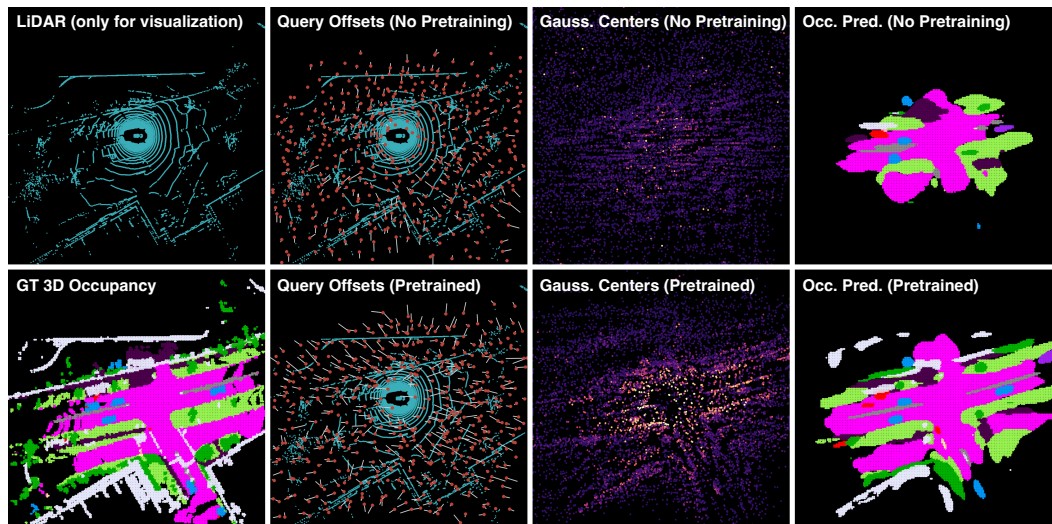

Figure 2: **Impact of denoising pretraining on occupancy estimation.** We visualize query offsets (column 2), Gaussian centers (column 3) *colored by opacity*, and occupancy estimation (column 4) for S2GO without and with pretraining. Without pre-training, queries remain largely stagnant, and Gaussians fail to capture 3D structures. In contrast, pretraining with rendering and denoising allows queries to move towards occupied regions—particularly visible for **walls** and **cars**—while Gaussians self-organize to better represent the scene, significantly improving occupancy estimation.

Gaussians are each locally pulled to different scene elements – resulting in suboptimal local minima (Kerbl et al., 2023; Charatan et al., 2024) – their corresponding queries are similarly stuck in suboptimal locations, unable to properly cover the scene.

### 3.3.2 DENOISING AND RENDERING FRAMEWORK

To explicitly supervise query movement and train Gaussians to model 3D geometry around their queries, we introduce a denoising and rendering framework for pretraining S2GO. The model functions as described in Section 3.2, but in this stage, we initialize current query locations $\mathbf{p}^i$ at noised LiDAR points at that timestep. Relaxing the $t$ subscript, given 3D points $\mathbf{pts} \in \mathbb{R}^{M \times 3}$, we set

$$\{\mathbf{p}^i\}_{i=0}^K = \text{FPS}_K(\mathbf{pts}) + \epsilon \tag{7}$$

where $m$ is the # of LiDAR points, $\epsilon \sim \text{U}(-e, e)^{K \times 3}$, $\text{FPS}_K$ applies Furthest-Point-Sampling (FPS) to yield $K$ points, and $\text{U}(-e, e)$ is the uniform distribution with $e$ as a hyperparameter. Starting at these noised positions, the model predicts query offsets $\{\mathbf{o}_t^i\}_{i=1}^K$ and derived Gaussians $\mathcal{G}_t$.

### 3.3.3 TRAINING OBJECTIVES

We then supervise these outputs with the loss function:

$$\mathcal{L} = \lambda_1 \sum_{i=1}^K ||\text{FPS}_K(\mathbf{pts_t}) - (\mathbf{p}_t^i + \mathbf{o}_t^i)|| + \lambda_2 \mathcal{L}_{depth}(\mathcal{G}, D) + \lambda_3 \mathcal{L}_{rgb}(\mathcal{G}, I) \tag{8}$$

The first term is the denoising objective, training the network to self-organize the queries to cover 3D structure. Then, $\mathcal{L}_{depth}$ and $\mathcal{L}_{rgb}$ render depth maps and RGB images from the Gaussians and supervise them with LiDAR projected depth maps $D_t$ and image observations. This explicitly trains Gaussians to represent detailed scene structure around the aligned queries. Notably, the rendering supervision is done on current and neighboring keyframes (+/- 0.5s) by moving the Gaussians with predicted velocities $v$ and accounting for ego-motion. This further improves final 3D occupancy performance. Altogether, this denoising and rendering stage provides S2GO with a strong prior for sparse queries and Gaussians to effectively model the 3D scene geometry.

## 3.4 STAGE 2: 3D SEMANTIC OCCUPANCY ESTIMATION

### 3.4.1 OCCUPANCY ESTIMATION FRAMEWORK

Equipped with the pretraining prior, S2GO is then trained for 3D semantic occupancy estimation. The model processes image observations, predicts a set of Gaussians $\mathcal{G}_t$ at each timestep, which now

also include semantic class predictions, and "splats" Gaussians to nearby voxels as in Section 3.1. Notably, unlike the pre-training phase, query positions are initialized at learnable 3D locations. As such, S2GO *only uses RGB images during inference*. The "splatted" voxel predictions are trained using ground truth semantic occupancy, and we additionally supervise neighboring frames similar to Stage 1. In this section, we propose crucial improvements to further strengthen this pipeline.

### 3.4.2 Opacity-Weighted Geometry Estimation

The Gaussian-to-voxel splatting framework in GaussianFormer-2 handles foreground classes as a mixture of Gaussians, and opacity is only used for weighting Gaussians inside the mixture. As such, opacity has no bearing on determining binary occupancy of a location, in contrast to Gaussians in rendering (Kerbl et al., 2023) where opacity acts as a proxy for density. This leads to unexpected behavior: Gaussians in unoccupied regions decrease their scale $\mathbf{s}$ and position themselves *between* voxel centers to minimize their occupancy contribution (Eq. 2), all while maintaining significant opacity. This unnatural representation for Gaussians conflicts with the rendering initialization and hurts performance. To address this issue, we additionally weight the occupancy probability $\alpha(\mathbf{x}; \mathbf{G})$ with the opacity estimation $a$, yielding:

$$\alpha(\mathbf{x}; \mathbf{G}) = a \exp\left(-\frac{1}{2}(\mathbf{x} - \mathbf{m})^{\mathrm{T}} \mathbf{\Sigma}^{-1}(\mathbf{x} - \mathbf{m})\right) \tag{9}$$

Our change significantly improves performance by encouraging Gaussians in unoccupied regions to simply predict lower opacity and stabilizing scale supervision to be more consistent.

### 3.4.3 Efficient Gaussian-to-Voxel Splatting

To implement Gaussian-to-voxel splatting, GaussianFormer (Huang et al., 2024) first determines pairs of interacting Gaussians and voxels, then parallelizes over voxels in the forward pass and over Gaussians in the backward pass. However, this formulation does not account for the inherent locality of the splatting operation: neighboring voxels process a similar set of Gaussians and vice-versa. Such voxels and Gaussians should be processed together in a CUDA block for optimized L1 cache usage. This is especially a problem for the backward pass since naively parallelizing over Gaussians incurs random access costs on a large number of voxels (640k).

To address this problem in the forward pass, we block voxels into 4x4x4 grids and have threads tied to each voxel collaboratively load nearby Gaussians onto memory before splatting them, similar to 3DGS (Kerbl et al., 2023). In the backward pass, we adopt a similar approach but additionally take care to tie threads to individual Gaussians to avoid atomic operations on the gradients (Mallick et al., 2024). Our efficient Gaussian-to-voxel splatting implementation, with 9k Gaussians and 640k voxels on an A100, speeds up the forward pass by **1.5×** (1.29ms to **0.87ms**) and the backward pass by **20.4×** (116ms to **5.7ms**), substantially reducing the wall-clock time required for training. Furthermore, it decreases the GPU memory cost of the forward/backwards pass by **3.3×** from 2013MB/2079MB to 631MB/633MB.

### 3.4.4 Query Propagation

A key point in our streaming 3D occupancy pipeline is query propagation. More specifically, we need to determine the optimal subset of current queries to push onto the queue for future timesteps. While a straightforward selection of top-k largest query opacities works, propagating the most confidently occupied regions of the scene, we find that queries end up highly overlapping over time, with insufficient coverage over the entire scene. To mitigate this, we choose the highest opacity queries that are pairwise separated by a distance $\delta$, where $\delta$ is a hyperparameter. This maintains an effective balance between maintaining high-opacity regions and distributing queries across the scene.

## 4 Experiments

We perform extensive experiments on three benchmarks derived from the nuScenes and KITTI datasets. S2GO uses the ResNet50 (He et al., 2016) backbone, S2GO-Small uses 900 queries with 10 Gaussians each, and S2GO-Base uses 1800 queries with 20 Gaussians. Additional details about the experiment setup are in Supplementary A and more implementation details are in Supplementary B.

### 4.1 Quantitative Results

We first evaluate S2GO on the nuScenes dataset. On the nuScenes-SurroundOcc benchmark in Table 1, S2GO-Small improves over the previous Gaussian-based state-of-the-art method Gaussian-

Table 1: **3D occupancy estimation results on the nuScenes-SurroundOcc validation set (Wei et al., 2023).** All baselines leverage a $900 \times 1600$ resolution while S2GO uses $256 \times 704$ resolution images for efficiency. All methods are benchmarked on the 4090 GPU. *GaussianWorld's paper results over-weight intermediate frames during evaluation. We re-evaluate released checkpoints under the standard setting.

| Method | IoU | mIoU | barrier | bicycle | bus | car | const. veh. | motorcycle | pedestrian | traffic cone | trailer | truck | drive. suf. | other flat | sidewalk | terrain | manmade | vegetation | FPS |
|---|---|---|---|---|---|---|---|---|---|---|---|---|---|---|---|---|---|---|---|
| MonoScene (Cao & de Charette, 2022) | 24.0 | 7.3 | 4.0 | 0.4 | 8.0 | 8.0 | 2.9 | 0.3 | 1.2 | 0.7 | 4.0 | 4.4 | 27.7 | 5.2 | 15.1 | 11.3 | 9.0 | 14.9 | - |
| Atlas (Murez et al., 2020) | 28.7 | 15.0 | 10.6 | 5.7 | 19.7 | 24.9 | 8.9 | 8.8 | 6.5 | 3.3 | 10.4 | 16.2 | 34.9 | 15.5 | 21.9 | 21.0 | 11.2 | 20.5 | - |
| BEVFormer (Li et al., 2022b) | 30.5 | 16.8 | 14.2 | 6.6 | 23.5 | 28.3 | 8.7 | 10.8 | 6.6 | 4.1 | 11.2 | 17.8 | 37.3 | 18.0 | 22.9 | 22.2 | 13.8 | 22.2 | 3.3 |
| TPVFormer (Huang et al., 2023) | 30.9 | 17.1 | 16.0 | 5.3 | 23.9 | 27.3 | 9.8 | 8.7 | 7.1 | 5.2 | 11.0 | 19.2 | 38.9 | 21.3 | 24.3 | 23.2 | 11.7 | 20.8 | 2.9 |
| OccFormer (Zhang et al., 2023) | 31.4 | 19.0 | 18.7 | 10.4 | 23.9 | 30.3 | 10.3 | 14.2 | 13.6 | 10.1 | 12.5 | 20.8 | 38.8 | 19.8 | 24.2 | 22.2 | 13.5 | 21.4 | - |
| SurroundOcc (Wei et al., 2023) | 31.5 | 20.3 | 20.6 | 11.7 | 28.1 | 30.9 | 10.7 | 15.1 | 14.1 | 12.1 | 14.4 | 22.3 | 37.3 | 23.7 | 24.5 | 22.8 | 14.9 | 21.9 | 3.3 |
| GaussianFormer (Huang et al., 2024) | 29.8 | 19.1 | 19.5 | 11.3 | 26.1 | 29.8 | 10.5 | 13.8 | 12.6 | 8.7 | 12.7 | 21.6 | 39.6 | 23.3 | 24.5 | 23.0 | 9.6 | 19.1 | 2.7 |
| GaussianFormer-2 (Huang et al., 2025) | 31.7 | 20.8 | 21.4 | 13.4 | 28.5 | 30.8 | 10.9 | 15.8 | 13.6 | 10.5 | 14.0 | 22.9 | 40.6 | 24.4 | 26.1 | 24.3 | 13.8 | 22.0 | 2.8 |
| QuadricFormer (Zuo et al., 2025a) | 31.2 | 20.1 | 19.6 | 13.1 | 27.3 | 29.6 | 11.3 | 16.3 | 12.7 | 9.2 | 12.5 | 21.2 | 40.2 | 24.3 | 25.7 | 24.2 | 13.0 | 21.9 | 6.2 |
| GaussianWorld* (Zuo et al., 2025b) | 32.8 | 21.8 | 21.6 | 13.3 | 27.3 | 31.2 | 13.9 | 16.9 | 13.3 | 11.8 | 14.8 | 23.7 | 41.9 | 24.3 | 28.4 | 26.3 | 15.7 | 24.5 | 4.4 |
| ALOcc-mini-GF (Chen et al., 2025a;b) | 34.6 | 23.1 | 22.2 | 16.0 | 27.9 | 32.7 | 12.1 | 18.9 | 16.6 | 15.3 | 14.5 | 23.9 | 46.0 | 28.2 | 29.0 | 26.6 | 15.8 | 23.7 | 5.4 |
| ALOcc-GF (Chen et al., 2025a;b) | 38.2 | 25.5 | 24.3 | 18.8 | 29.8 | 34.3 | 17.9 | 19.6 | 17.5 | 17.2 | 15.5 | 26.5 | 47.6 | 29.9 | 31.2 | 29.2 | 20.0 | 29.0 | 0.9 |
| S2GO-Small | 34.3 | 22.1 | 20.8 | 13.1 | 27.5 | 30.3 | 14.5 | 16.5 | 11.7 | 10.9 | 13.5 | 23.3 | 46.3 | 29.2 | 29.7 | 28.4 | 13.0 | 25.1 | **26.1** |
| S2GO-Base | 35.5 | 22.7 | 21.9 | 13.4 | 27.5 | 32.1 | 14.9 | 15.3 | 12.9 | 11.8 | 13.4 | 24.0 | 46.9 | 29.1 | 30.3 | 29.1 | 14.7 | 26.4 | 19.6 |

Table 2: **Results on the SSCBench-KITTI-360 test set (Geiger et al., 2012) with a monocular camera.** S2GO achieves new state-of-the-art, achieving strong performance in all categories.

| Method | Input | IoU | mIoU | car | bicycle | motorcycle | truck | other-veh. | person | road | parking | sidewalk | other-grnd | building | fence | vegetation | terrain | pole | traf.-sign | other-struct. | other-object |
|---|---|---|---|---|---|---|---|---|---|---|---|---|---|---|---|---|---|---|---|---|---|
| LMSCNet (Roldao et al., 2020) | L | 47.5 | 13.7 | 20.9 | 0.0 | 0.0 | 0.3 | 0.0 | 0.0 | 63.0 | 13.5 | 33.5 | 0.2 | 43.7 | 0.3 | 40.0 | 26.8 | 0.0 | 0.0 | 3.6 | 0.0 |
| SSCNet (Song et al., 2017) | L | 53.6 | 17.0 | 32.0 | 0.0 | 0.2 | 10.3 | 0.6 | 0.1 | 65.7 | 17.3 | 41.2 | 3.2 | 44.4 | 6.8 | 43.7 | 28.9 | 0.8 | 0.8 | 8.6 | 0.7 |
| MonoScene (Cao & de Charette, 2022) | C | 37.9 | 12.3 | 19.3 | 0.4 | 0.6 | 8.0 | 2.0 | 0.9 | 48.4 | 11.4 | 28.1 | 3.2 | 32.9 | 3.5 | 26.2 | 16.8 | 6.9 | 5.7 | 4.2 | 3.1 |
| Voxformer (Li et al., 2023a) | C | 38.8 | 11.9 | 17.8 | 1.2 | 0.9 | 4.6 | 2.1 | 1.6 | 47.0 | 9.7 | 27.2 | 2.9 | 31.2 | 5.0 | 29.0 | 14.7 | 6.5 | 6.9 | 3.8 | 2.4 |
| TPVFormer (Huang et al., 2023) | C | 40.2 | 13.6 | 21.6 | 1.1 | 1.4 | 8.1 | 2.6 | 2.4 | 53.0 | 12.0 | 31.1 | 3.8 | 34.8 | 4.8 | 30.1 | 17.5 | 7.5 | 5.9 | 5.5 | 2.7 |
| OccFormer (Zhang et al., 2023) | C | 40.3 | 13.8 | 22.6 | 0.7 | 0.3 | 9.9 | 3.8 | 2.8 | 54.3 | 13.4 | 31.5 | 3.6 | 36.4 | 4.8 | 31.0 | 19.5 | 7.8 | 8.5 | 7.0 | 4.6 |
| GaussianFormer (Huang et al., 2024) | C | 35.4 | 12.9 | 18.9 | 1.0 | 4.6 | 18.1 | 7.6 | 3.4 | 45.5 | 10.9 | 25.0 | 5.3 | 28.4 | 5.7 | 29.5 | 8.6 | 3.0 | 2.3 | 9.5 | 5.1 |
| GaussianFormer-2 (Huang et al., 2025) | C | 38.4 | 13.9 | 21.1 | 2.6 | 4.2 | 12.4 | 5.7 | 1.6 | 54.1 | 11.0 | 32.3 | 3.3 | 32.0 | 5.0 | 28.9 | 17.3 | 3.6 | 5.5 | 5.9 | 3.5 |
| S2GO-Base (ours) | C | **40.8** | **15.1** | 22.7 | 1.3 | 1.7 | 15.9 | 5.1 | 2.1 | 53.8 | 13.3 | 33.4 | 3.8 | 35.3 | 7.2 | 31.2 | 21.1 | 6.4 | 6.5 | 6.0 | 4.2 |

World (Zuo et al., 2025b) by 1.5 IoU while substantially improving inference speed by **5.9×**. The larger S2GO-Base further improves IoU by 2.7 and retains a 4.5× speed advantage, demonstrating the efficacy of our sparse, query-based framework. We note that S2GO achieves these performance improvements while using a *smaller* image size (900x1600 vs 256x704). Additional experiments with equivalent image size are in Appendix H.

Compared with the state-of-the-art grid-aligned method Chen et al. (2025b), S2GO-Base achieves comparable results to ALOcc-mini-GF while being 3.6× faster in FPS. While ALOcc-GF achieves the highest mIoU on SurroundOcc, its throughput is 0.9 FPS, placing it outside the real-time regime. In contrast, our S2GO-Small and S2GO-Base variants run at 19.0 and 23.6 FPS, respectively, and are specifically designed for high-throughput deployment, offering a more favorable accuracy–efficiency trade-off in this real-time regime. We also provide results on nuScenes-Occ3D in Supplementary J, where we achieve competitive results with high FPS.

We also evaluate our approach on the KITTI-360 dataset (Geiger et al., 2012) in Table 2. In this monocular 3D semantic occupancy estimation setting, S2GO again achieves state-of-the-art performance, substantially improving over GaussianFormer-2 (Huang et al., 2025).

## 4.2 QUALITATIVE ANALYSIS

In Fig. 3, we qualitatively compare GaussianWorld and our method, visualizing different timesteps from two example driving sequences in nuScenes. After several seconds, after both the ego vehicle and surrounding vehicles move, GaussianWorld struggles to maintain independent representations of distinct objects and *incorrectly merges multiple instances*. This limitation occurs because GaussianWorld, despite its streaming nature, directly operates on low-level Gaussian primitives without object-level representations. Consequently, its local convolutions merge nearby objects. In contrast,

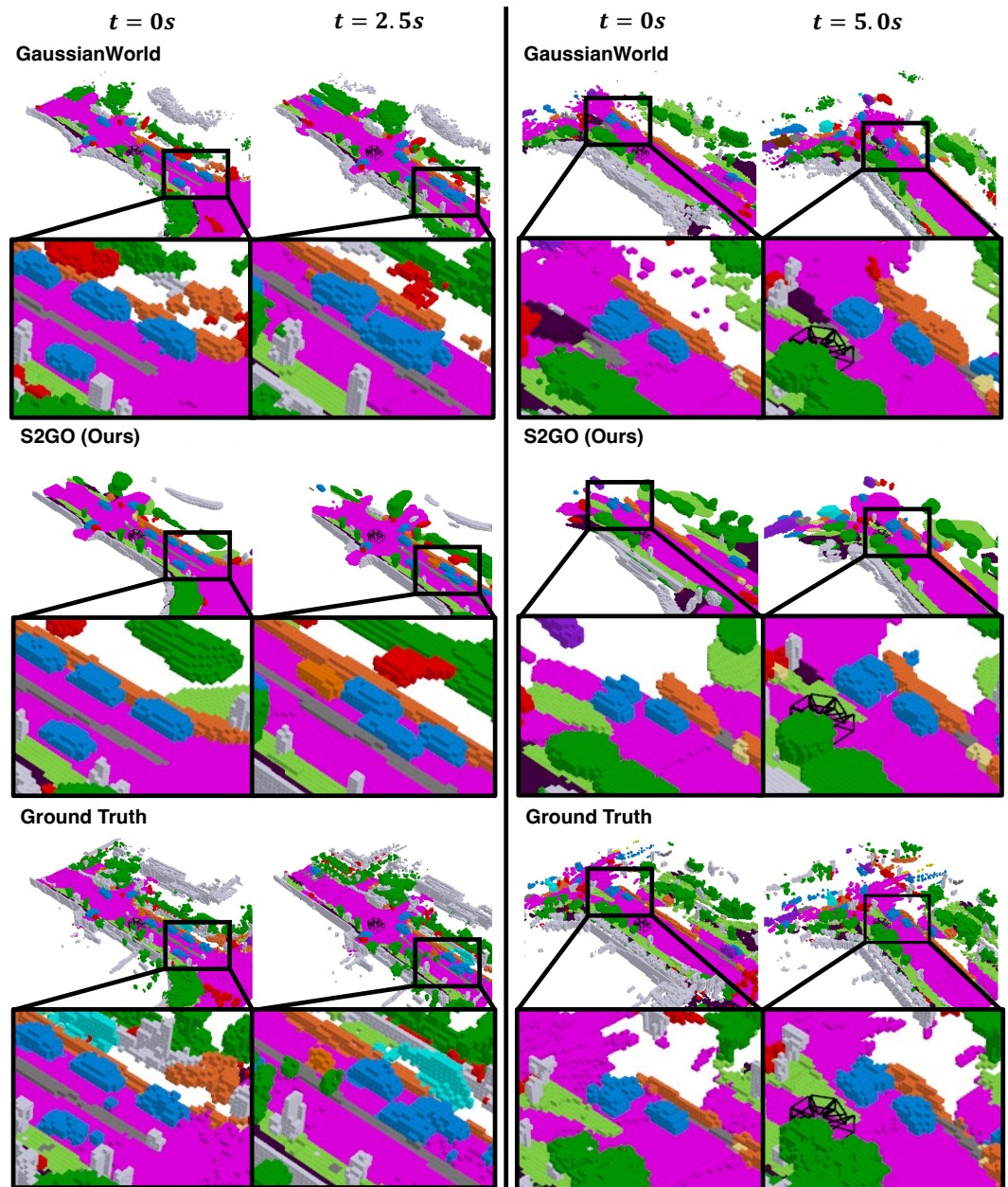

Figure 3: **Qualitative comparison of occupancy estimation.** We compare S2GO with Gaussian-World (Zuo et al., 2025b) by visualizing different timesteps from two example driving sequences. GaussianWorld incorrectly merges different instances over time, while S2GO effectively preserves distinct object identities by operating at a higher semantic level with sparse queries.

S2GO hierarchically decomposes the scene into a sparse set of queries with constituent Gaussians, enabling it to operate at a higher semantic level and effectively preserve distinct object identities. We provide qualitative results on SSCBench-KITTI-360 in Supplementary K.

## 4.3 ABLATIONS

In this section, we verify the effectiveness of our proposed components. By default, models are trained for 12 epochs during both stages. All ablations are on the SurroundOcc-nuScenes dataset.

**Pretraining.** In Table 3, we ablate the impact of pretraining. First, directly training occupancy estimation for 12 epochs (a) or 24 epochs (a)† yields poor results due to ambiguous supervision.

Table 3: **Ablation on pretraining.** We ablate *pretraining* query initialization and the depth, RGB, and query denoising loss terms. LiDAR + $\epsilon$ indicates queries are initialized from noised LiDAR. Pretraining with all objectives is essential. [†]Trained for 24 epochs.

| | Query Init | Depth | RGB | Denoise | mIoU | IoU |
|---|---|---|---|---|---|---|
| (a) | - | ✗ | ✗ | ✗ | 13.02 | 25.73 |
| (a)[†] | - | ✗ | ✗ | ✗ | 15.83 | 28.35 |
| (b) | Learnable | ✓ | ✓ | ✗ | 12.42 | 26.64 |
| (c) | LiDAR | ✓ | ✓ | ✗ | 13.62 | 27.08 |
| (d) | LiDAR+$\epsilon$ | ✓ | ✓ | ✗ | 20.55 | 32.68 |
| (e) | LiDAR+$\epsilon$ | ✓ | ✗ | ✗ | 20.25 | 32.44 |
| (d) | LiDAR+$\epsilon$ | ✓ | ✓ | ✗ | 20.55 | 32.68 |
| (f) | LiDAR+$\epsilon$ | ✓ | ✓ | ✓ | **21.60** | **33.91** |

Table 4: **Ablation on query propagation strategies.** "None" indicates no temporal information is used.

| Propagation Type | mIoU | IoU |
|---|---|---|
| None | 17.92 | 29.24 |
| top-k opacity | 19.94 | 32.03 |
| $\delta$-dist top-k opacity | **20.51** | **32.51** |

Table 5: **Ablation on using velocity modeling in each stage.**

| Pretrain. | Occ. Est. | mIoU | IoU |
|---|---|---|---|
| ✗ | ✗ | 20.07 | 31.87 |
| ✗ | ✓ | 20.15 | 31.94 |
| ✓ | ✗ | 20.50 | 32.62 |
| ✓ | ✓ | **20.55** | **32.68** |

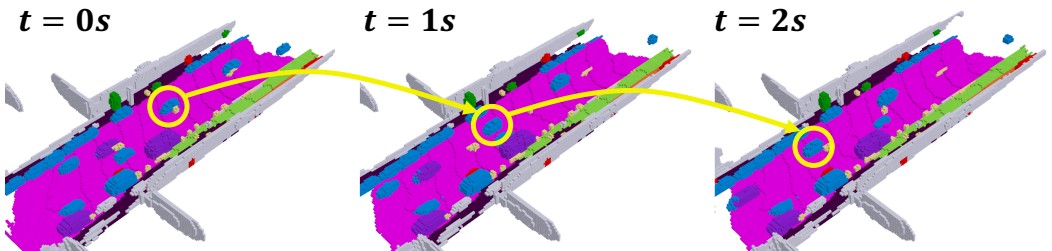

Figure 4: **Visualization of future occupancy predictions.** We use the self-supervised velocity prediction for each query to roll out future occupancy predictions. Our streaming query-based framework well-decouples motion of individual objects.

Next, we pretrain with depth and RGB supervision and ablate query initialization. Learnable initialization during pretraining – which is what S2GO uses in the second stage – is *worse* than not pretraining. This occurs because such queries are randomly distributed over 3D space, with most queries far from occupied geometry and hence unable to get adequate supervision. On the other hand, initializing query locations precisely at LiDAR points is only slightly better than not pretraining – this baseline supervises Gaussians to capture local geometry, but the queries themselves are not supervised to move. Finally, adding noise to LiDAR before initializing achieves remarkable performance, providing meaningful supervision to both queries and Gaussians. We emphasize that this is the only initialization method that substantially improves over not pretraining with the same compute budget (24 epochs of occupancy in (a)† vs 12+12 epochs with pretraining).

Next, we ablate each pretraining loss function. Depth supervision alone is enough to achieve good performance. Adding RGB loss slightly boosts results as RGB supervises finer details, and denoising supervision gives a substantial final boost. This confirms that our proposed pretraining is essential for our streaming, sparse-query framework to reach its potential and achieve state-of-the-art.

**Query Propagation.** Query selection for future frames is critical for streaming perception. In Table 4, we ablate propagation strategies. Compared to the single-frame baseline without propagation, selecting top-k queries by opacity already provides a substantial performance gain. However, this leads to excessive overlap between queries over time, wasting capacity in the model. Enforcing a minimum distance between queries enables wider query coverage, further improving performance. Supplementary D further examines S2GO performance with varying history propagation horizon.

**Velocity Modeling.** S2GO predicts a velocity for each query, which is used in both stages to move dynamic regions before applying RGBD or occupancy supervision in neighboring frames. While this module is useful on its own for future occupancy prediction as shown in Fig. 4, we ablate its impact on performance in Table 5. Velocity modeling improves performance in both stages, with motion modeling during pretraining proving particularly important.

**Gaussian to Voxel Splatting.** In Table 6, we ablate our inclusion of opacity $a$ in occupancy probability $\alpha$ and our efficient Gaussian-to-voxel splatting algorithm. First, including opacity substantially improves performance

Table 6: **Ablation on Gaussian-to-Voxel Splatting (G2V).** GPU Training hours are calculated for training 12 epochs on a single GPU. All benchmarks are calculated on a single 4090 GPU.

| Opacity in $\alpha$ | Efficient G2V | mIoU | IoU | Train GPU hours | Infer GPU Mem. | Infer. FPS |
|---|---|---|---|---|---|---|
| ✗ | ✗ | 16.97 | 28.75 | 55h | 2436 MB | 25.2 |
| ✓ | ✗ | 20.13 | 32.28 | 129h | 7043 MB | 20.4 |
| ✓ | ✓ | 20.55 | 32.68 | 28h | 2448 MB | 25.3 |

(+3.16 mIoU), but doubles the training time as Gaussians opt to reduce occupancy probability by lowering opacity instead of scale, thus increasing the number of voxels each Gaussian affects. We resolve this by using our optimized CUDA kernels, which further improves performance while substantially lowering training cost, *halving* the original training time. Furthermore, our optimized kernels also substantially reduce the required GPU memory for single-batch inference from 7099 MB to 2511 MB, and also improve inference throughput from 24.1 frames/s to 31.5 frames/s.

**Number of Gaussians.** In Table 7 we ablate the # of queries and Gaussians. We observe that even just 900 sparse queries and 10 Gaussians per query is enable to capture the overall scene and achieve a high mIoU with a real-time 26.1 FPS on a 4090. With more queries and Gaussians, the performance steadily improves, but at the cost of longer runtime.

**Pretraining with Various Sources of Depth.** While our use of LiDAR during pretraining does not add data collection overhead, since LiDAR is already required to generate occupancy labels, we also investigate pretraining on different sources of depth in Table 7. In case LiDAR is not available during training, replacing LiDAR with occupied voxel locations yields almost identical results. Replacing the nuScenes' 32-line LiDAR with a cheaper 16-line sensor largely preserves performance, and replacing LiDAR entirely with zero-shot monocular depth predictions from Metric3D on RGB images also achieves good results. This demonstrates the general applicability of our pretraining pipeline.

Table 7: **Ablation on the number of queries and Gaussians.** FPS is measured on a 4090 GPU.

| # Query | # Gauss./Query | # Gauss. | mIoU | IoU | FPS |
|---|---|---|---|---|---|
| 900 | 10 | 9000 | 21.60 | 33.91 | **26.1** |
| 1260 | 14 | 17640 | 21.78 | 34.15 | 22.7 |
| 1800 | 20 | 36000 | **21.84** | **34.51** | 19.6 |

Table 8: **Ablation on pretraining with various depth sources.**

| Pretraining Query Initialization | mIoU | IoU |
|---|---|---|
| Occupied voxel locations | 21.61 | 33.75 |
| LiDAR (32-line) | 21.60 | 33.91 |
| LiDAR (16-line) | 21.16 | 33.39 |
| Zero-shot RGB depth pred. (Yin et al., 2023) | 20.99 | 33.57 |

## 5 CONCLUSION

We presented a novel framework for 3D semantic occupancy estimation that leverages sparse 3D queries to efficiently capture and propagate scene information over time. Our method replaces traditional dense, grid-aligned Gaussian representations with a compact, streaming set of semantic queries. A geometry denoising pre-training phase ensures effective alignment of sparse queries with dense occupancy targets, accurately modeling both static and dynamic scene elements. Extensive evaluations on nuScenes and KITTI benchmarks demonstrate state-of-the-art performance while operating 5.9× faster than previous methods. Our work demonstrates that a query-based approach can effectively bridge the gap between efficiency and high-fidelity 3D scene representation. In the future, we plan to explore multitask, end-to-end learning and large-scale pretraining using unlabeled data to further enhance model performance and generalization.

**Reproducibility Statement.** To aid reproducibility, we have provided details throughout the paper regarding our S2GO framework as well as our efficient Gaussian-to-voxel splatting algorithm. Section 3.2 provides general details regarding the architecture, and Supplementary B provides low-level details including image size, learning rate, channel dimension, and other hyperparameters. Dataset and evaluation details are in Supplementary A, and details regarding the splatting algorithm are in Section 3.4.3.

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

# A   ADDITIONAL DETAILS ON THE EXPERIMENT SETUP

**Datasets.** We conducted comprehensive experiments on three benchmarks derived from nuScenes and KITTI. The *nuScenes dataset* (Caesar et al., 2020) provides 1000 scenes of surround-view driving scenes. We evaluate our method on both the SurroundOcc (Wei et al., 2023) and Occ3D (Tian et al., 2023) benchmarks. SurroundOcc provides voxel-based annotations in a $100 \times 100 \times 8$ m² range around the car with a $200 \times 200 \times 16$ resolution, classifying voxels into 18 classes (16 semantic, 1 empty, and 1 noise). Occ3D offers voxelized semantic occupancy in a $80 \times 80 \times 6.4$ m² range with a $200 \times 200 \times 16$ resolution, derived from an auto-labeling pipeline. The *KITTI dataset* (Geiger et al., 2012) comprises over 320k images and 80k laser scans from suburban driving scenes. We adopt the dense semantic annotations from SSCBench-KITTI-360 (Li et al., 2024; Liao et al., 2022b). The official split consists of 7/1/1 sequences for training, validation, and testing, respectively. The voxel grid spans an area of $51.2 \times 51.2 \times 6.4$ m² in front of the ego car, with a resolution of $256 \times 256 \times 32$. Each voxel is classified into one of 19 classes (18 semantic categories and 1 empty).

**Evaluation Metrics.** Following MonoScene (Cao & de Charette, 2022), we use **IoU** and **mIoU** as evaluation metrics. For the Occ3D dataset, we adopt RayIoU as our primary metric following SparseOcc (Liu et al., 2024), **RayIoU** extends mIoU by evaluating occupancy estimation at the ray level rather than voxel level. It simulates LiDAR rays and assesses predictions based on both depth accuracy and class correctness. RayIoU ensures balanced evaluation by resampling rays across distances and incorporating temporal casting from past, present, or future viewpoints to assess scene completion. By preventing inflated IoU scores caused by thick surface predictions and applying a depth threshold for true positive classification, RayIoU provides a more robust evaluation. Metrics are defined as:

$$\text{mIoU/RayIoU} = \frac{1}{|C|} \sum_{i \in C} \frac{TP_i}{TP_i + FP_i + FN_i} \tag{10}$$

$$\text{IoU} = \frac{TP_{\neq c_0}}{TP_{\neq c_0} + FP_{\neq c_0} + FN_{\neq c_0}} \tag{11}$$

where $TP_i$, $FP_i$, and $FN_i$ are the number of true positive, false positive, and false negative predictions for class $i$, $C$ is the set of semantic classes, and $c_0$ is the nonempty class. For RayIoU, a query ray is classified as a true positive (TP) if the predicted class matches the ground truth and the L1 error between the predicted and ground-truth depth is within a certain threshold (1m, 2m, 4m)

**Baselines** We evaluate S2GO against representative approaches spanning diverse 3D representation paradigms. Specifically, we compare with voxel-based methods, including MonoScene (Cao & de Charette, 2022), Atlas (Murez et al., 2020), SurroundOcc (Wei et al., 2023), which employ dense 3D voxel grids for occupancy reconstruction. We further benchmark against BEV-based methods like BEVFormer (Li et al., 2022b). In addition, we consider the triplane-based TPVFormer (Huang et al., 2023), which decomposes 3D space into orthogonal 2D planes, facilitating efficient feature aggregation. Lastly, we include Gaussian-based approaches—GaussianFormer (Huang et al., 2024), GaussianFormer-2 (Huang et al., 2025), and GaussianWorld (Zuo et al., 2025b)—which employ 3D Gaussians to model 3D occupancy and semantics.

# B   ADDITIONAL IMPLEMENTATION DETAILS

On nuScenes, S2GO uses a 256x704 resolution image and is pre-trained on denoising and rendering for 12 epochs without semantic annotations, and then trained for 24 epochs for 3D semantic occupancy estimation. S2GO-Small uses an ImageNet1k backbone, while S2GO-Base leverages nuImages pre-training. On KITTI, we use a 256x1408 resolution image and an ImageNet1k backbone. The model is pre-trained for 12 epochs, then trained for occupancy for another 12 epochs.

The temporal transformer closely follows the design from PETR (Liu et al., 2022) and StreamPETR (Wang et al., 2023), with a 4-frame (2 second) queue. Each transformer block consists of a self-attention layer across all the queries, followed by a cross-attention layer to refine queries based on image features, and a feedforward layer. The self attention layer includes keys and queries derived from past queries as in StreamPETR, and we use deformable attention for the cross attention layer for efficiency. All models are trained with a 4e-4 learning rate with a batch size of 16, with the cosine annealing schedule and the AdamW optimizer with weight decay 0.01. We use gradient clipping with max norm 35, and scale the learning rate for the backbone by a factor 0.25. All of our models are trained with mixed precision. On nuScenes-SurroundOcc, the LiDAR nosing factor $\epsilon$ is set to 1 meter. During training, the pairwise query distance $\delta$ for query propagation is randomly sampled between 0 to 3 meters, and during inference, it is set to 1.6m. For nuScenes-Occ3D and KITTI, all distances are scaled according to the smaller extent of the 3D scene. The embedding dimension of the temporal transformer is 768, and we leverage Flash Attention (Dao et al., 2022) for efficient self-attention between queries. Queries interact with the image through Deformable Attention (Zhu et al., 2020; Lin et al., 2022; Wang et al., 2023).

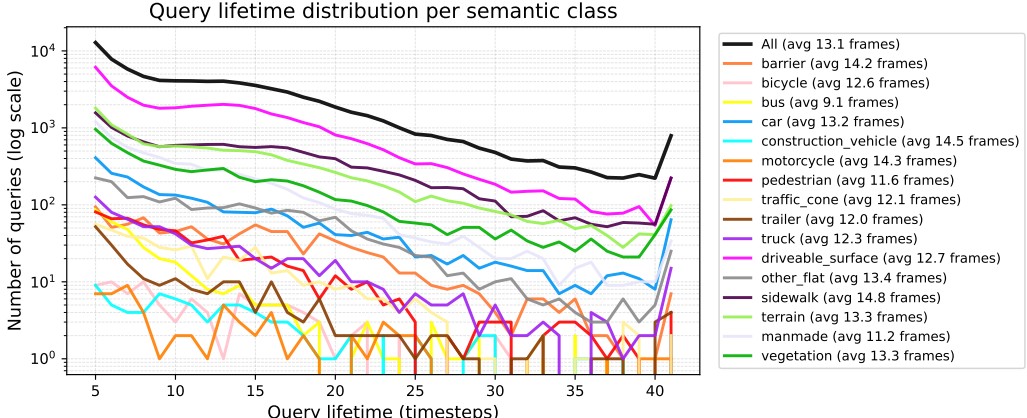

Figure 5: **Distribution of query lifetimes per semantic class.** Most queries continue propagating over a significant number of frames, demonstrating that our queries are persistent over time.

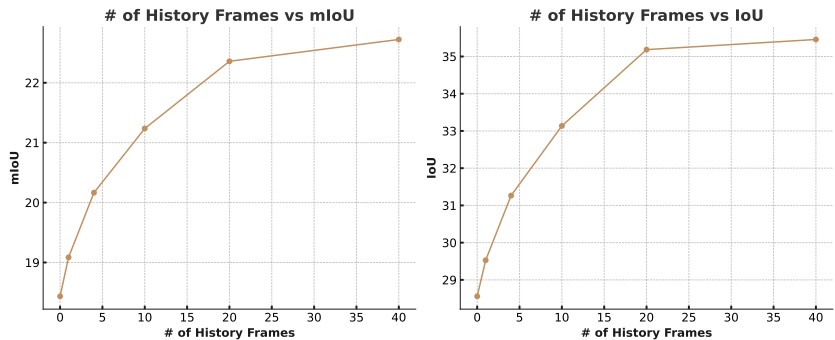

Figure 6: **Impact of history length on occupancy performance.** A longer history consistently improves performance, showcasing the advantage of our streaming approach over prior projection-based methods.

## C ANALYSIS ON QUERY PERSISTENCE

In Figure 5, we analyze query persistence over time. For each query, we measure how long it remains in the active query set after first being spawned, and we plot the resulting lifetime distribution as well as per-class averages. Following tracking works, we prune short-lived tracklets during this analysis. We find that a substantial fraction of queries persist for tens of steps ($>$5 seconds), demonstrating that once a query explains a region or object, it remains active throughout much of the sequence. Taken together with Figure 4, this analysis demonstrates that S2GO can effectively handle static and dynamic elements of the scene with persistent queries.

## D ABLATION ON THE NUMBER OF HISTORY FRAMES

To further evaluate S2GO, we plot occupancy performance over different streaming history lengths in Figure 6. With a longer history, performance steadily improves, demonstrating the efficacy of our streaming framework. We emphasize that unlike prior projection-based works, S2GO incurs *no additional cost* from a longer history.

## E ANALYSIS ON THE QUEUE LENGTH VS QUERY DISTANCE $\delta$

In Figure 7, we examine how queue length and query distance $\delta$ affect the mIoU and IoU of S2GO-Base. First, we notice that performance steadily improves with longer history, also as seen in Figure 6. Next, we find that a query distance $\delta$ of around 1.5m is best for most history lengths.

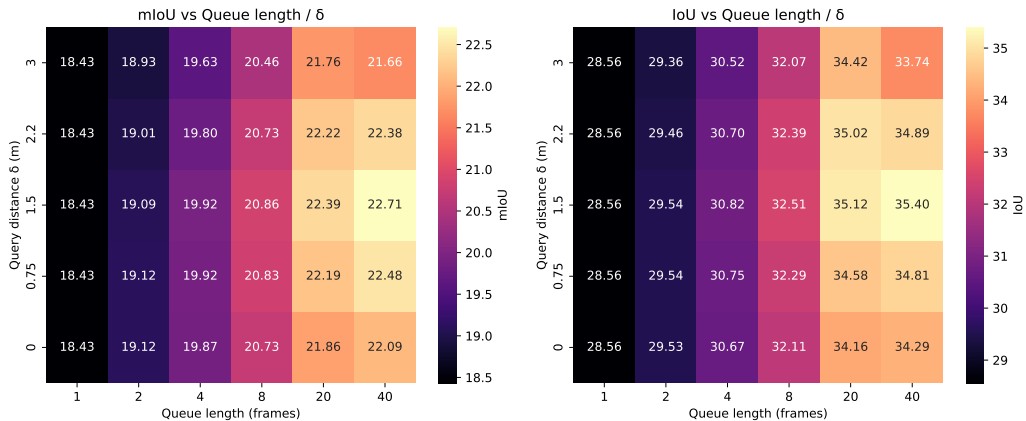

Figure 7: **Analysis of the relationship between the queue length and query distance $\delta$.** A queue length of 1 indicates no history is used.

Notably, we find that with shorter history, the query distance $\delta$ matters less. For instance, the difference in IoU between the best and worst $\delta$ for a long history of 40 frames is 1.66 IoU. However, for a history length of 4, the difference is just 0.3 IoU. This indicates that maintaining a diverse set of spread-out queries is increasingly important the longer the sequence proceeds. This is because unlike other methods that repeatedly sample past image features, in S2GO, the query set is the only source of history information passed onto the future. If queries are poorly selected, the past information is lost.

## F    ANALYSIS ON THE SCALE $s$ VS OPACITY $a$ FOR OCCUPIED & UNOCCUPIED VOXELS

In this section, we thoroughly investigate the impact of including opacity $a$ into the occupancy probability in the Gaussian-to-voxel splatting formulation. In Section 3.4.2, we proposed that GaussianFormer2's Huang et al. (2025) the exclusion of opacity $a$ in the occupancy probability is unnatural, as then opacity has no bearing on determining binary occupancy of a location.

In Figure 8 (top), we visualize the 2D heatmap of learned pairs of 3D Gaussian sizes and opacities $a$ for GaussianFormer2. We use the $1\sigma$ 3D Gaussian ellipsoid volume computed from the scales $s$ as the 3D Gaussian size. We find that while GaussianFormer2 uses a broad range Gaussian sizes to represent the scene as expected, the opacities for occupied regions are **lower** than the opacities for unoccupied regions. This is because in GaussianFormer-2, opacity is only used as the weight in the mixture-of-Gaussians for determining foreground classes. As a result, Gaussians in unoccupied regions are forced to learn to position themselves *between voxel centers* instead of simply lowering their opacity.

We visualize this clearly in Figure 9, where we plot the distance between predicted 3D Gaussian centers and the closest voxel center for both occupied (blue) and unoccupied (orange) regions. We clearly see a huge distribution difference in distances for GaussianFormer2, confirming our hypothesis. This Gaussian formulation destabilizes training and is suboptimal: slight shifts in the Gaussian locations causes unwanted foreground artefacts to appear in the background.

To address this, we propose in S2GO to weight the occupancy probability by opacity $a$. This change results to a more natural formulation of occupancy, where opacity dynamically adapts to explain regions in foreground, as shown in Figure 6 (bottom-left). Furthermore, the network easily learns to predict low opacity for background regions (Figure 6, bottom-right) and is not forced to learn to position Gaussians away from voxel centers for unoccupied regions (Figure 7, left).

## G    DETAILED LATENCY BREAKDOWN

We benchmark our 9000 Gaussian model on an A100 GPU. The backbone, temporal transformer, gaussian estimation, and propagation take 11.54ms, 22.79ms, 2.22ms, and 1.45ms, respectively.

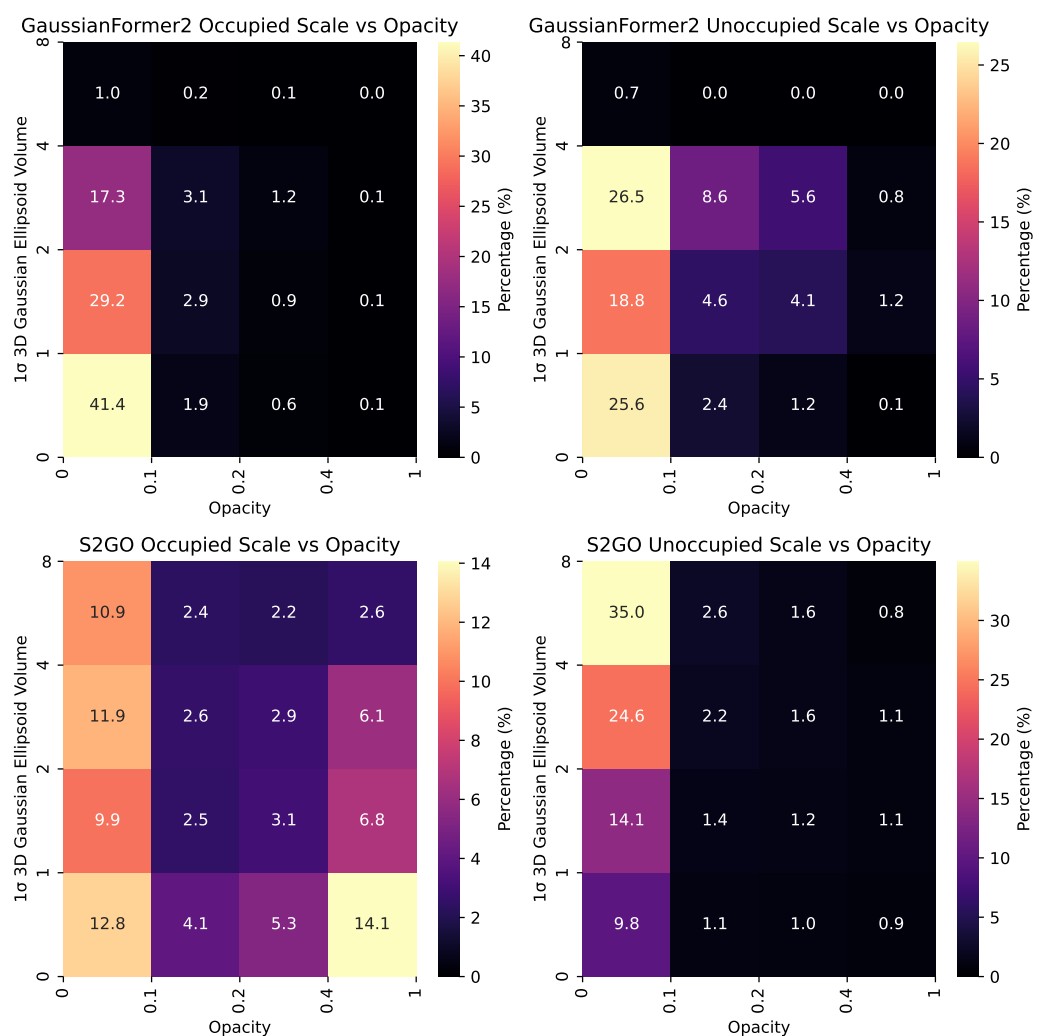

Figure 8: **Analysis of learned 3D Gaussian size vs opacity $a$ over occupied and unoccupied voxels for GaussianFormer2 vs S2GO.**

Table 9: Quantitative comparisons with QuadricFormer Zuo et al. (2025a) with 256x704 resolution. All methods are benchmarked on an **A6000 GPU**.

| Method | IoU | mIoU | barrier | bicycle | bus | car | const. veh. | motorcycle | pedestrian | traffic cone | trailer | truck | drive. suf. | other flat | sidewalk | terrain | manmade | vegetation | FPS |
|---|---|---|---|---|---|---|---|---|---|---|---|---|---|---|---|---|---|---|---|
| QuadricFormer-3.2k | 28.9 | 18.5 | 18.0 | 11.0 | 25.7 | 28.0 | 11.0 | 13.3 | 11.7 | 9.1 | 11.5 | 19.2 | 39.0 | 23.1 | 24.5 | 22.9 | 9.4 | 19.2 | 22.3 |
| QuadricFormer-12.8k | 30.3 | 18.9 | 18.2 | 10.6 | 25.3 | 28.4 | 11.0 | 12.6 | 11.6 | 9.4 | 12.3 | 19.8 | 39.9 | 22.6 | 25.6 | 23.7 | 10.8 | 19.8 | 13.9 |
| S2GO-Small | 34.3 | 22.1 | 20.8 | 13.1 | 27.5 | 30.3 | 14.5 | 16.5 | 11.7 | 10.9 | 13.5 | 23.3 | 46.3 | 29.2 | 29.7 | 28.4 | 13.0 | 25.1 | **32.2** |
| S2GO-Base | **35.5** | **22.7** | 21.9 | 13.4 | 27.5 | 32.1 | 14.9 | 15.3 | 12.9 | 11.8 | 13.4 | 24.0 | 46.9 | 29.1 | 30.3 | 29.1 | 14.7 | 26.4 | 23.6 |

# H ADDITIONAL COMPARISONS ON NUSCENES

We additionally compare with the recent method QuadricFormer Zuo et al. (2025a) with the same 256x704 resolution image input into the same ResNet-50 backbone in Table 9. We find that S2GO consistently demonstrates stronger performance at the same, or faster, FPS. For instance, QuadricFormer-3.2k and S2GO-Base have similar FPS, but S2GO-Base delivers a 6.6 higher IoU and 4.2 higher mIoU.

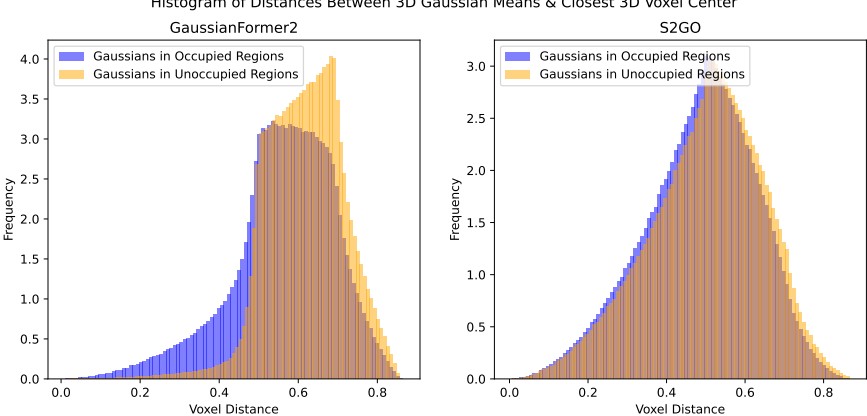

Figure 9: **Histogram of distances between 3D Gaussian centers and the nearest voxel center.**

Further, while we were unable to develop a high-resolution ($900 \times 1600$) version of the model due to computational limitations, we have benchmarked its FPS for reference. The S2GO-Base model with high resolution settings as in QuadricFormer achieves 130.2ms, reaching 7.7 FPS. We note that in such a model, the backbone alone occupies most of the computation, and the difference in efficiency enabled by our sparse query framework is most apparent when the backbone is comparatively lightweight in real-time inference settings.

# I    COMPARISON OF GROUND TRUTH IN SURROUNDOCC AND OCC3D

In Figure 10 we present side-by-side comparisons of ground truth in SurroundOcc-nuScenes and Occ3D-nuScenes. While both accurately capture the underlying scene, we find that SurroundOcc often generates more complete ground truth labels along with a larger scene extent ($100m \times 100m$ vs $80m \times 80m$). While both datasets are strong testbeds for semantic occupancy development, in this work, we focus on the SurroundOcc-nuScenes benchmark as it enables longer-range perception and serves as the main testbed for Gaussian-focused methods.

# J    ADDITIONAL NUSCENES-OCC3D EVALUATION RESULTS

While our experiments are focused on nuScenes-SurroundOcc, which provides more complete ground truth, we also provide extensive comparisons with existing methods on the nuScenes-Occ3D benchmark using detailed metrics, as shown in Tab. 10. Our method achieves strong RayIoU while maintaining a fast inference speed, demonstrating the potential of our sparse, query-based occupancy framework.

# K    QUALITATIVE RESULTS ON SSCBENCH-KITTI-360

In Fig. 11 we visualize example predictions and ground truth from the SSCBench-KITTI-360 dataset. Our framework flexible adapts to a monocular setting and precisely predicts the semantic occupancy of the driving scene.

# L    LLM USAGE.

An LLM was lightly used at the end for checking grammar and re-wording small parts of the manuscript.

SurroundOcc GT         Occ3D GT

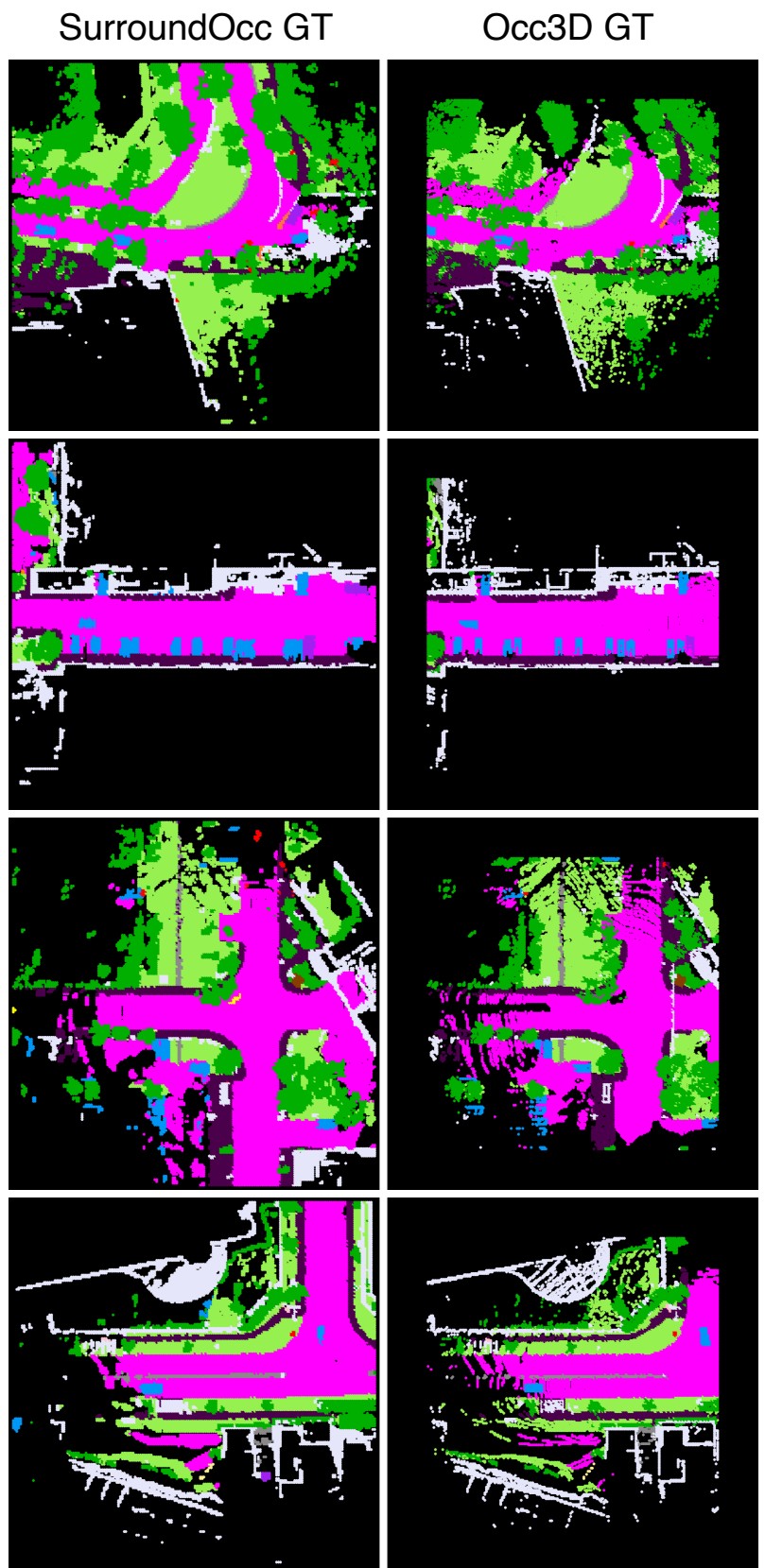

Figure 10: **Comparison of 3D GT generated by SurroundOcc-nuScenes and Occ3D-nuScenes.**

Table 10: **3D occupancy performance on the Occ3D-nuScenes validation set** (Tian et al., 2023). FPS is benchmarked on both an A100 and on a 4090. A100 numbers for prior work are sourced from SparseOcc (Liu et al., 2024) and original papers, while 4090 numbers are sourced from ALOcc (Chen et al., 2025a). [†]ProtoOcc benchmarks on a 3090 GPU.

| Method | Backbone | Mask | Input Size | Epoch | RayIoU | RayIoU$_{1m, 2m, 4m}$ | | | mIoU | FPS$^{A100}$ | FPS$^{4090}$ |
|---|---|---|---|---|---|---|---|---|---|---|---|
| BEVFormer (Li et al., 2022b) | R101 | ✓ | 1600×900 | 24 | 32.4 | 26.1 | 32.9 | 38.0 | 39.2 | 3.0 | - |
| RenderOcc (Pan et al., 2024) | Swin-B | ✓ | 1408×512 | 12 | 19.5 | 13.4 | 19.6 | 25.5 | 24.4 | - | - |
| SimpleOcc (Gan et al., 2024) | R101 | ✓ | 672×336 | 12 | 22.5 | 17.0 | 22.7 | 27.9 | 31.8 | 9.7 | - |
| BEVDet-Occ (Huang & Huang, 2021) | R50 | ✓ | 704×256 | 90 | 29.6 | 23.6 | 30.0 | 35.1 | 36.1 | 2.6 | - |
| BEVDet-Occ-Long (Huang & Huang, 2021) | R50 | ✓ | 704×384 | 90 | 32.6 | 26.6 | 33.1 | 38.2 | 39.3 | 0.8 | - |
| FB-Occ (Li et al., 2023b) | R50 | ✓ | 704×256 | 90 | 33.5 | 26.7 | 34.1 | 39.7 | 39.1 | 10.3 | - |
| ProtoOcc (Kim et al., 2025) | R50 | ✓ | 704×256 | 24 | - | - | - | - | 39.6 | - | 12.8[†] |
| ProtoOcc (Oh et al., 2025) | R50 | ✓ | 800×432 | 12 | - | - | - | - | 39.0 | - | - |
| BEVFormer (Li et al., 2022b) | R101 | ✗ | 1600×900 | 24 | 33.7 | - | - | - | 23.7 | 3.0 | 4.4 |
| FB-Occ (Liu et al., 2024) | R50 | ✗ | 704×256 | 90 | 35.6 | - | - | - | 27.9 | 10.3 | - |
| SparseOcc (Liu et al., 2024) | R50 | ✗ | 704×256 | 48 | 36.1 | 30.2 | 36.8 | 41.2 | 30.9 | 12.5 | - |
| GSD-Occ (He et al., 2025) | R50 | ✗ | 704×256 | 24 | 38.9 | - | - | - | - | 20.0 | - |
| StreamOcc (Moon et al., 2025) | R50 | ✗ | 704×256 | 24 | 41.1 | 34.2 | 41.9 | 47.1 | - | 12.0 | - |
| OPUS-L (Wang et al., 2024a) | R50 | ✗ | 704×256 | 100 | 41.2 | 34.7 | 42.1 | 46.7 | 36.2 | 7.2 | 8.2 |
| STCOcc (Liao et al., 2025) | R50 | ✗ | 704×256 | 36 | 42.1 | 36.9 | 42.8 | 46.7 | - | - | - |
| ALOcc-3D (Chen et al., 2025a) | R50 | ✗ | 704×256 | 54 | 43.7 | 37.8 | 44.7 | 48.8 | 38.0 | - | 6.0 |
| ALOcc-GF (Chen et al., 2025b) | R50 | ✗ | 704×256 | 24 | **44.1** | 38.2 | 45.0 | 49.2 | - | - | 6.2 |
| S2GO-Small (ours) | R50 | ✗ | 704×256 | 24 | 37.2 | 31.3 | 38.1 | 42.2 | 30.8 | 20.8 | 25.3 |
| S2GO-Base (ours) | R50 | ✗ | 704×256 | 24 | 39.1 | 33.1 | 40.0 | 44.1 | 31.2 | 14.5 | 20.9 |

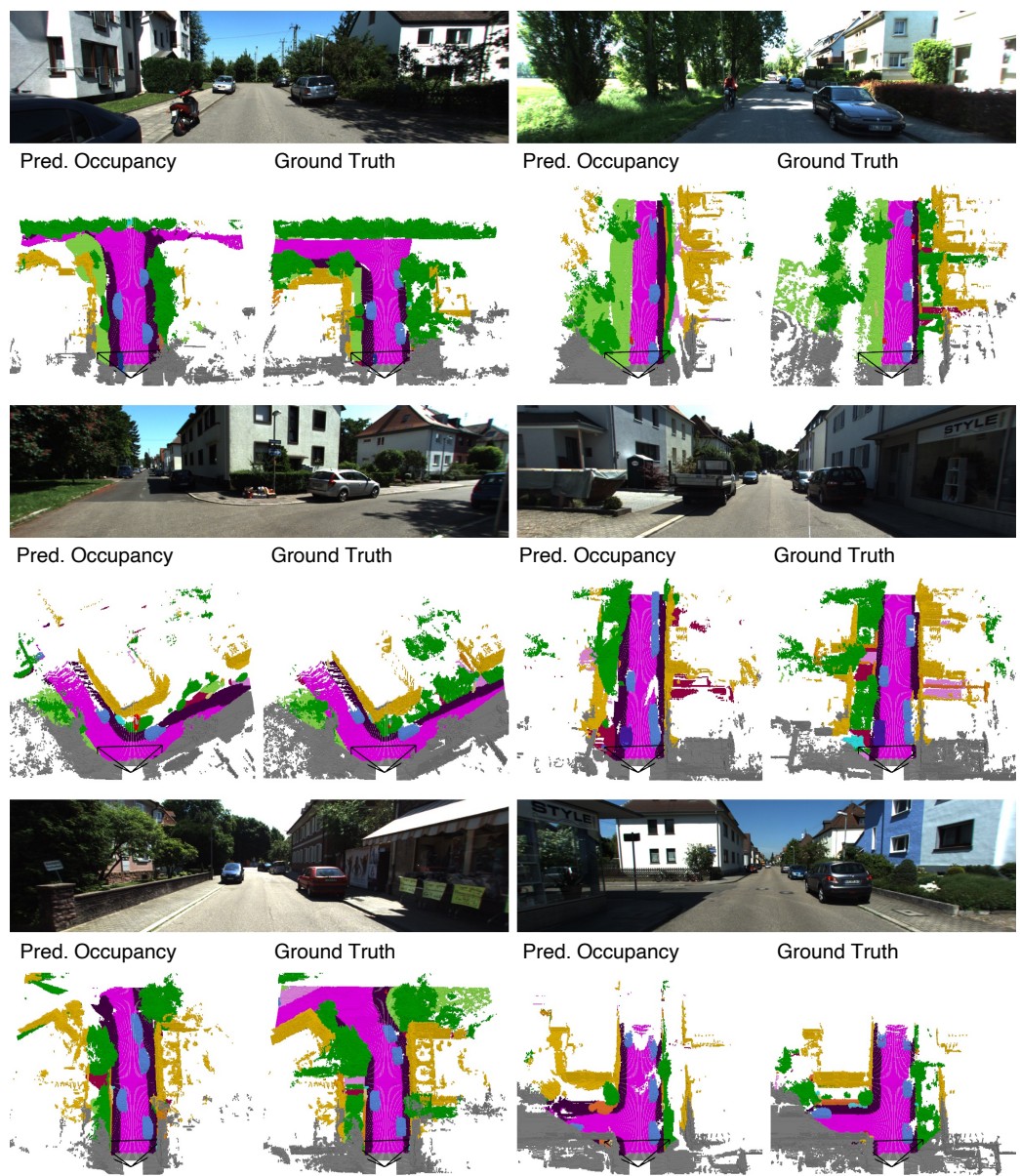

Figure 11: **Qualitative Results on the SSCBench-KITTI-360 dataset.** S2GO well-captures occupancy details even in a monocular setting.

