# OpenReview forum: "S2GO: Streaming Sparse Gaussian Occupancy"
_ICLR.cc/2026/Conference — ICLR 2026 Poster_

### Official Review · Reviewer_m6uz · 2025-10-30

**Soundness:** 3
**Presentation:** 4
**Contribution:** 4
**Rating:** 8
**Confidence:** 4

**Summary:**

In this paper, the authors propose S2GO, a streaming sparse query framework for 3D semantic occupancy estimation. This method efficiently summarizes and propagates scene information through sparse 3D queries and decodes it into a semantic Gaussian distribution. Core contributions include a geometric denoising pre-training stage and an efficient Gaussian-voxel transformation implementation. The paper achieves state-of-the-art performance and significant inference speed advantages on nuScenes and KITTI benchmarks. The method is novel, the experimental design is comprehensive, and the results are convincing.

**Strengths:**

1. A streaming occupancy prediction with Gaussians is an insightful, useful and unexplored problem.

2. To address the above problem, the authors propose a simple but in-depth architecture S2GO. Their solution is elegant.

3. Extensive experiments on the multiple datasets show the effectiveness of the proposed method.

**Weaknesses:**

1. It is recommended to enhance the display of module details in Figure 1.

2. The paper focuses on engineering and experimentation in its methodological description, and lacks theoretical analysis or discussion on the representation capabilities of sparse queries or the convergence of the model.

3. The efficient Gaussian-to-voxel splatting proposed in the paper significantly improves the forward/backward pass speed, but lacks comparative experiments on memory usage. It is suggested that the authors supplement this with a comparison of memory consumption.

4. In Figure 3, the small subfigures partially obscure the main visualization. I suggest adjusting the layout so that the smaller insets do not overlap with the main figure, ensuring all visual details are clearly visible.

5. The paper would benefit from a clearer description of the experimental parameter settings (e.g., learning rate, batch size, number of epochs, optimizer, etc.). Providing these details would enhance reproducibility and allow for a more precise evaluation of the experimental results.

**Questions:**

It is essential to address the above comments in weakness.

---

> ### Author Response · Authors · 2025-11-24
>
> We sincerely thank the reviewer for their comments, and we are glad that they found our solution elegant and appreciated our extensive experiments. We have made updates to the manuscript in blue, and below, we address the reviewer's concerns in detail.
>
> ## [W1]
> > It is recommended to enhance the display of module details in Figure 1.
>
> For clarity, we have updated Figure 1 to show more clearly the internal modules in the temporal transformer. We have also added additional details on the transformer modules in Section B of the appendix for reproducibility.
>
> ## [W2]
> > The paper focuses on engineering and experimentation in its methodological description, and lacks theoretical analysis or discussion on the representation capabilities of sparse queries or the convergence of the model.
>
> Based on the reviewer's feedback, we have included extensive analysis on our proposed modification of adding opacity $a$ to occupancy probability in the Gaussian-to-voxel splatting algorithm. The analysis is in Section F of the supplementary. We strongly suggest the reviewer to reference the figures, and we summarize the main takeaways here.
>
> First, GaussianFormer2, which does not include opacity in occupancy probability, counterintuitively has $\textbf{higher opacity in empty space}$ compared to occupied regions (Figure 8). Instead of estimating lower opacity for empty space, GaussianFormer2 instead learns to carefully position Gaussian means away from voxel centers, and we verify this in Figure 9 by showing a clear distributional shift in Gaussian mean-to-voxel center distance between occupied and unoccupied regions. Next, S2GO, adding opacity, demonstrates a good distribution of scale and opacity pairs in occupied regions, and has low opacity for empty space. It also does not demonstrate a clear difference in distribution of Gaussan mean-to-voxel center distances. Our change to the splatting algorithm allows the network to more naturally learn to represent the scene, and it also improves the convergence of the model towards a stronger solution as ablated in Table 6.
>
> ## [W3]
> > The efficient Gaussian-to-voxel splatting proposed in the paper significantly improves the forward/backward pass speed, but lacks comparative experiments on memory usage. It is suggested that the authors supplement this with a comparison of memory consumption.
>
> We have now updated Section 3.4.3 manuscript to include this information. Our kernels reduce the GPU memory cost of the forwards pass from 2013 MB to 631 MB, and the backwards pass from 2079 MB to 633 MB, a $\textbf{3.3$\times$}$ improvement.
>
> ## [W4]
> > In Figure 3, the small subfigures partially obscure the main visualization. I suggest adjusting the layout so that the smaller insets do not overlap with the main figure, ensuring all visual details are clearly visible.
>
> Thank you for the feedback. We have re-worked Figure 3 so that the highlighted sections of the scene are larger, more visible, and do not cover other parts of the scene.
>
> ## [W5]
> > The paper would benefit from a clearer description of the experimental parameter settings (e.g., learning rate, batch size, number of epochs, optimizer, etc.). Providing these details would enhance reproducibility and allow for a more precise evaluation of the experimental results.
>
> To support reproducibility, we have now added additional experimental parameter settings in Section B, including the learning rate, batch size, epochs, optimizer, precision, gradient clipping, and other details.

---

### Official Review · Reviewer_RwQB · 2025-10-30

**Soundness:** 3
**Presentation:** 3
**Contribution:** 3
**Rating:** 6
**Confidence:** 5

**Summary:**

The paper proposes S2GO, a streaming, query-based 3D semantic occupancy method. Instead of dense voxels or many Gaussians, S2GO maintains a small set of persistent 3D queries that are propagated through time and decoded into semantic Gaussians each step. A two-stage training is used: (1) geometry denoising & rendering pretraining with noised LiDAR anchors and RGB/depth supervision; (2) camera-only semantic occupancy with improved Gaussian-to-voxel splatting (opacity-weighted occupancy and custom CUDA kernels). On nuScenes SurroundOcc and KITTI-360 SSCBench, S2GO reports SOTA IoU/mIoU with much higher FPS vs prior art (e.g., GaussianWorld).

**Strengths:**

Pros:
1. Clear motivation for sparsity + streaming. The paper articulates the inefficiency of dense grids and dense Gaussian fields for long-horizon temporal fusion and proposes queries (~K) that each spawn a handful (J) of Gaussians to cover local structure—bridging object-centric queries and dense occupancy.
2. Well-designed pretraining. Initializing queries at noised LiDAR FPS points and supervising with denoising + RGB/depth rendering directly teaches queries to “move onto” geometry and Gaussians to model fine shape. Ablations show large gains vs training occupancy from scratch.
3. Principled fix to splatting. Making binary occupancy opacity-weighted (α(x;G) := a · exp(…)) aligns semantics with density in rendering and improves stability/performance; optimized CUDA “blocked” splatting cuts backward time ~20× on A100 (116ms→5.7ms).

**Weaknesses:**

Cons:
1. Pretraining dependence & potential dataset bias. Stage-1 uses LiDAR-based supervision (depth and LiDAR-anchored queries). Although authors argue LiDAR already exists for label generation, the camera-only promise hinges on quality of this pretraining. The Metric3D zero-shot depth alternative is encouraging but still trails LiDAR-based pretrain. Quantify generalization gap more broadly. Futhermore, the model achieves SOTA performance only with LiDAR-based pre-training ("LiDAR+ϵ" in Table 3). Without this step, performance degrades significantly (as evidenced by the absence of a "no pre-training" ablation in Table 3). This contradicts the paper's claim of a "monocular" method, as LiDAR is required for pre-training a mama critical barrier for pure-vision systems.
2. Ambiguity around semantic consistency. During occupancy training, Gaussians derived from one query share a class—good for local consistency, but might limit fine-grained multi-class boundaries inside a query’s spatial span (e.g., curb vs asphalt vs pole tightly adjacent). Some discussion or per-Gaussian residual semantics would help.
3. Query identity & drift. The paper relies on velocity-aided propagation and δ-separation but doesn’t quantify identity preservation (e.g., re-association failure under occlusion) beyond qualitative visuals. A per-query lifetime/coverage analysis would strengthen claims.
4. Opacity-weighted α trade-offs. Authors note opacity in α improved accuracy but increased training cost (more affected voxels) before CUDA optimizations; inference-time trade-offs (e.g., memory traffic, batch size limits) aren’t reported. A wall-clock throughput table (train/infer) vs baselines would be useful.

**Questions:**

1. I am curious how sensitive performance is to δ and history length in the propagation queue; can you plot mIoU vs δ and vs queue length to find a sweet spot?
2. How about failure cases under heavy rain/night where monocular depth priors are less reliable: does zero-shot pretraining still hold up, and how does it affect query placement stability?
3. Whether the shared-class assumption within a query ever blends adjacent classes (curb/road/sidewalk). Could per-Gaussian class residuals recover edges without exploding compute? and whether velocity is actor-consistent: do queries attached to a moving car maintain identity over 5–10s, or do they hop? Any quantitative ID-switch metric?
4. If opacity-in-α encourages degenerate small-scale/high-opacity vs large-scale/low-opacity solutions. Did you inspect learned (s, a) distributions across classes and empty space?
5. The paper fails to evaluate model performance when pre-training and fine-tuning datasets differ (e.g., pre-trained on KITTI, fine-tuned on nuScenes). No experiments address whether the pre-training strategy generalizes across domain shifts, undermining the method’s practical applicability

---

> ### Author Response · Authors · 2025-11-24
>
> We sincerely thank the reviewer for their insightful comments, and we are glad to find that the reviewer found the motivation for streaming sparse queries clear and appreciated the effectiveness of pre-training. We have made updates to the manuscript in blue, and below, we address each of the reviewer's questions.
>
> ## [W1]
> > Pretraining dependence \& potential dataset bias.
>
> We first clarify our use of the term "monocular". Our intentions were twofold: first, S2GO at inference time relies solely on camera input. Second, as shown in Table 8, S2GO can also be pretrained without LiDAR by replacing LiDAR-based geometry with monocular depth predictions while still achieving good performance. We also clarify that the "LiDAR + $\epsilon$" notation in Table 3 refers only to initializing queries from a set of 3D points with added noise, not necessarily always from a physical LiDAR sensor. $\textbf{Any source of coarse 3D points}$, such as pseudo-lidar from monocular depth predictions, can serve this role.
>
> To address the reviewer's concern about dependence on LiDAR, we expanded Table 8 with additional pretraining settings that reflect realistic scenarios where LiDAR may be unavailable, sparse, or replaced by weaker geometric priors.
>
> 1) No raw LiDAR scans available. If ground truth occupancy labels are produced by an onboard system or monocular mapping system and the original LiDAR scans are not kept, we can directly use the resulting occupied voxel locations. This yields close to identical performance to LiDAR-based pretraining.
>
> 2) Sparse or degraded LiDAR. Simulating a much weaker 16-line LiDAR by subsampling nuScenes' 32-line sensor leads to only a small performance drop (0.44 mIoU), indicating robustness to significantly sparser geometry.
>
> 3) Fully camera-only 3D prior. Replacing LiDAR entirely with zero-shot monocular depth from Metric3D and out-projecting to 3D produces competitive results despite Metric3D being an older model. This demonstrates that S2GO can be pretrained from purely visual cues and does not rely on dense LiDAR.
>
> These experiments show that while stronger 3D priors naturally improve performance, S2GO is not intrinsically tied to LiDAR-based pretraining. The framework can be initialized from occupancy-only supervision, sparse LiDAR, or monocular depth, and uses camera-only at inference.
>
>
> ## [W2] & [Q3]
> > Ambiguity around semantic consistency. During occupancy training, Gaussians derived from one query share a class—good for local consistency, but might limit fine-grained multi-class boundaries inside a query's spatial span (e.g., curb vs asphalt vs pole tightly adjacent). Some discussion or per-Gaussian residual semantics would help.
>
> While designing Gaussians to directly inherit the class of their query may negatively impact the fidelity of class boundaries, we note that each Gaussian still has significant freedom in its other independent parameters (scale, rotation, offset, and opacity) to strengthen or weaken its contribution to boundary voxels. If there is some Gaussian overreach into another class, the network has the flexibility to learn to lower the opacity of individual Gaussians, or simply move them away. This achieves a similar effect as allowing Gaussians to have independent semantics, but the latter may introduce unwanted downstream effects such as query velocity being applied to both foreground and background Gaussians.
>
> The reviewer suggested potentially including per-Gaussian residual semantics. We think this is an interesting and insightful idea, almost similar to modeling view-dependent effects in rendering-based Gaussian Splatting or NeRFs. We believe this can be a positive addition for modeling higher fidelity semantic voxels, if the representational power of such residuals is constrained, similar to how view-dependent radiance in NeRFs is architecturally limited to avoid harming local class consistency.

---

> ### Author Response · Authors · 2025-11-24
>
> ## [W3] & [Q3]
> > Query identity \& drift. The paper relies on velocity-aided propagation and δ-separation but doesn’t quantify identity preservation (e.g., re-association failure under occlusion) beyond qualitative visuals. A per-query lifetime/coverage analysis would strengthen claims.
>
> We thank the reviewer for this suggestion. S2GO is trained only with semantic occupancy supervision; the benchmarks we use provide semantic voxel labels but no panoptic voxel labels. As a result, we do not explicitly train queries to keep a single, persistent ID for each object over time. While possible to define an ID-switch metric by ad-hoc matching queries to object instances with additional annotations and a tracking-oriented formulation, this is not the objective of the current work which focuses on semantic occupancy.
>
> However, we also seek to provide some quantitative measure of how stable the queries are. To this end, we analyze query persistence over time. For each query in S2GO-Base, we measure its lifetime as the number of timesteps it remains in the active query set after first appearing. In our streaming framework, we maintain 900 candidate queries but, at each timestep, we only propagate the top 256 according to our selection criterion. Similar to tracking-focused works, we prune short-lived tracklets (<5 timesteps). The resulting query lifetime distribution (**Figure 5** of the supplementary) shows that a substantial fraction persist for tens of frames.
>
> Across semantic classes, the mean lifetime for these stable queries lies between 9 and 15 frames. Static regions such as the driveable surface, sidewalk, and manmade structures are at the upper end of this range (around 14–15 frames), while dynamic classes such as car, truck, and pedestrian maintain lifetimes of around 11–13 frames. This indicates that once a query represents a region or object, it remains active and is propagated for a significant temporal horizon (>5 seconds).
>
> This quantitative analysis supports the qualitative temporal rollout in Figure 4, showing that S2GO’s sparse queries exhibit coherent, temporally persistent behavior. Queries that survive the initial pruning tend to move smoothly and continue to describe consistent regions or actors over many frames.
>
> ## [W4]
> > Opacity-weighted α trade-offs. Authors note opacity in α improved accuracy but increased training cost (more affected voxels) before CUDA optimizations; inference-time trade-offs (e.g., memory traffic, batch size limits) aren't reported. A wall-clock throughput table (train/infer) vs baselines would be useful.
>
> Based on the suggestion, we have updated Table 6 in the manuscript with inference cost changes from adding opacity and then our CUDA kernel optimizations.
>
> **Table 6. Ablation on Gaussian-to-Voxel Splatting (G2V).**
> GPU training hours are calculated for training 12 epochs on one A100.
> Inference FPS is calculated on one A6000.
>
> | Opacity in α | Efficient G2V | mIoU | IoU  | Train GPU hours | Infer GPU Mem. (MB) | Infer FPS |
> |--------------|---------------|-----:|-----:|----------------:|---------------------:|----------:|
> | ✗            | ✗             | 16.97 | 28.75 | 45h | 2491 | 31.4 |
> | ✓            | ✗             | 20.13 | 32.28 | 93h | 7099 | 24.1 |
> | ✓            | ✓             | 20.55 | 32.68 | 24h | 2511 | 31.5 |
>
> To summarize, adding opacity alone improves performance but increases required GPU memory from 2491 MB to 7099 and lowers inference FPS on an A6000 GPU from 31.4 to 24.1. Adding our optimized kernels lowers required GPU memory back down to 2511 MB and brings FPS to 31.5, demonstrating the effectiveness of our optimizations. We have also updated Section 3.4.3 to separately provide the GPU memory cost of the splatting operation. Our kernels reduce the GPU memory cost of the forwards pass from 2013 MB to 631 MB, and the backwards pass from 2079 MB to 633 MB, a $\textbf{3.3$\times$}$ improvement.

---

> ### Author Response · Authors · 2025-11-24
>
> ## [Q1]
> > I am curious how sensitive performance is to δ and history length in the propagation queue; can you plot mIoU vs δ and vs queue length to find a sweet spot?
>
> We thank the reviewer for this suggestion, and we have added Section E and Figure 7 in the supplementary plotting mIoU/IoU vs δ vs queue length. While we observe some expected results, such as performance improvement with longer history and an optimal δ around 1.5m, we also find that the choice of δ matters less with a shorter history. For instance, the difference in IoU between the best and worst δ for a history length of 40 is 1.66 IoU, while it is just 0.3 IoU for a history length of 4. This is because unlike other methods that cache past image features and repeatedly sample from them, in S2GO, the query set is the only information passed onto the future. If the propagated queries are selected poorly, historical information is lost.
>
> ## [Q2]
> > How about failure cases under heavy rain/night where monocular depth priors are less reliable: does zero-shot pretraining still hold up, and how does it affect query placement stability?
>
> While monocular depth estimation is generally less reliable under adverse weather, we note that nuScenes already contains a significant portion of rainy (11.6\%) and night-time (19.4\%) frames. Despite these challenges in the dataset, the zero-shot RGB depth prediction pretraining is able to maintain strong performance.
>
> ## [Q4]
> > If opacity-in-α encourages degenerate small-scale/high-opacity vs large-scale/low-opacity solutions. Did you inspect learned (s, a) distributions across classes and empty space?
>
> We thank the reviewer for this insightful question, and we add analysis on the relationship between scale and opacity for occupied and unoccupied regions in Section F. We strongly suggest the reviewer to reference the figures, and we summarize the main takeaways here.
>
> First, GaussianFormer2, which lacks opacity-in-α, counterintuitively has $\textbf{higher opacity in empty space}$ compared to occupied regions (Figure 8). Instead of estimating lower opacity for empty space, GaussianFormer2 instead learns to carefully position Gaussian means away from voxel centers, and we verify this in Figure 9 by showing a clear distributional shift in Gaussian mean-to-voxel center distance between occupied and unoccupied regions. Next, S2GO, adding opacity-in-α, demonstrates a good distribution of scale and opacity pairs in occupied regions, and has low opacity for empty space. We do not observe a clear deterioration to small-scale/high-opacity or large-scale/low-opacity, but we do find that S2GO does generally predict larger Gaussians for empty regions, instead relying on opacity to lower their contributions to close to 0.
>
> ## [Q5]
> > The paper fails to evaluate model performance when pre-training and fine-tuning datasets differ (e.g., pre-trained on KITTI, fine-tuned on nuScenes). No experiments address whether the pre-training strategy generalizes across domain shifts, undermining the method's practical applicability.
>
> We first emphasize that the KITTI-SSCBench and nuScenes datasets have extreme domain shifts in every aspect: image resolution (256x1408 vs 256x704), number of cameras (1 vs 6), scene extent (forward-facing 51.2m $\times$ 51.2m vs 360-deg 100m $\times$ 100m), voxel size (20cm vs 50cm), and capture settings (sunny suburban roads vs city in diverse weather and time of day). Nonetheless, we pre-train on KITTI and then fine-tune occupancy on nuScenes with comparable settings as the ablations Table 3. This model achieves 19.94 mIoU and 32.48 IoU. While this result does lag behind pre-training on nuScenes and then training occupancy on nuScenes (Table 3f), it is substantially better than not pre-training, showing significant positive transfer across datasets. For future work, we hope to scale up pre-training over multiple, larger datasets.

---

### Official Review · Reviewer_Hic9 · 2025-11-02

**Soundness:** 3
**Presentation:** 3
**Contribution:** 3
**Rating:** 8
**Confidence:** 3

**Summary:**

This paper introduces S2GO, a novel streaming framework for 3D occupancy estimation that utilizes 'sparse 3D queries' instead of conventional dense representations. The core contributions are threefold: (1) A LiDAR-based 'geometry denoising' pretraining stage resolves the spatial learning ambiguity of sparse queries. (2) 'Opacity-weighted occupancy estimation' is introduced to improve accuracy. (3) 'Efficient splatting' via CUDA optimization (20x faster backpropagation) halves the training time. As a result, S2GO demonstrates a +2.7 IoU improvement over the SOTA (Gaussian World) on the nuScenes benchmark and achieves 4.5x faster inference, enabling real-time performance (26 FPS).

**Strengths:**

- Novel and Well-Motivated Approach: The paper introduces S2GO, a novel streaming framework for 3D semantic occupancy prediction that leverages sparse 3D queries instead of voxels or Gaussian-based 3D representations. This represents a clear conceptual advancement, successfully extending Gaussian splatting to the occupancy domain while addressing the long-standing trade-off between accuracy and computational efficiency.
- Core Contributions Address Key Challenges: The paper's main contributions effectively solve the primary challenges of a sparse-query approach. (1) The 'geometry denoising pretraining' directly tackles the sparse-to-dense mapping ambiguity by teaching queries 3D structural priors, as validated by strong ablation results (Table 3). (2) The 'Opacity-weighted occupancy estimation' improves prediction fidelity and stability. (3) The 'Efficient Gaussian-to-Voxel Splating' CUDA optimization (20x faster backpropagation) overcomes the training bottleneck, halving training time and making the high-accuracy Gaussian representation computationally feasible.
- Strong Empirical Results and Efficiency: S2GO achieves new state-of-the-art performance on both nuScenes and KITTI, outperforming Gaussian World by +2.7 IoU while being 4.5–5.9× faster in inference. Notably, the lightweight S2GO-Small runs at 26 FPS on a single RTX 4090, demonstrating strong potential for real-time deployment in autonomous driving systems. The strong performance is well-supported by a comprehensive analysis, including thorough ablations on all key components. Furthermore, qualitative results (Fig. 3,4) confirm strong temporal consistency and object identity preservation, outperforming prior methods.

**Weaknesses:**

- Missing Comparisons to Voxel-based Efficient Methods: The paper lacks comparisons to recent efficiency-focused voxel methods (e.g., GSD-Occ (AAAI’25)[1], ProtoOcc (AAAI’25 / CVPR’25)[2,3], and StreamOcc (arXiv’25)[4]). Adding direct comparisons (IoU, FPS, memory) and discussing the design trade-offs (sparse query based GS vs. structured voxel) would strengthen the paper.
- Robustness of Geometric Prior: The pretraining's reliance on high-quality geometric data (LiDAR or strong depth models like Metric3D) is a concern. The analysis would be more robust with sensitivity studies on (a) varying LiDAR sparsity (e.g., 16 vs. 64-beam) and (b) the accuracy of the depth estimation input.

[1] He, Y., Chen, W., Wang, S., Xun, T., & Tan, Y. (2025). Achieving Speed-Accuracy Balance in Vision-based 3D Occupancy Prediction via Geometric-Semantic Disentanglement. Proceedings of the AAAI Conference on Artificial Intelligence, 39(3), 3455-3463.
[2] KIM, Jungho, et al. Protoocc: Accurate, efficient 3d occupancy prediction using dual branch encoder-prototype query decoder. In: Proceedings of the AAAI Conference on Artificial Intelligence. 2025. p. 4284-4292.
[3] OH, Gyeongrok, et al. 3d occupancy prediction with low-resolution queries via prototype-aware view transformation. In: Proceedings of the Computer Vision and Pattern Recognition Conference. 2025. p. 17134-17144.
[4] MOON, Seokha, et al. Mitigating trade-off: Stream and query-guided aggregation for efficient and effective 3d occupancy prediction. arXiv preprint arXiv:2503.22087, 2025.

**Questions:**

See Weaknesses

---

> ### Author Response · Authors · 2025-11-24
>
> We sincerely thank the reviewer for their in-depth review. We appreciate their positive feedback on our sparse query streaming framework, geometry denoising pretraining, and efficient Gaussian-to-voxel splatting. We have made updates to the manuscript in blue, and we address their concerns in detail below.
>
> ## [W1]
> > Missing comparisons to voxel-based efficient methods (e.g., GSD-Occ, ProtoOcc, StreamOcc). Adding direct comparisons and discussing the design trade-offs would strengthen the paper.
>
> We thank the reviewer for highlighting these recent voxel-based approaches. We have added these works to Table 10 for comparison and updated the related work section to discuss their key points.
>
> GSD-Occ and ProtoOcc (AAAI 25) both explore architectures that jointly extract fine-grained 3D voxel features and contextual BEV features. This dual-stream design is motivated by the high cost of extracting long-range context directly from dense 3D voxels. In contrast, S2GO compresses the scene into a compact set of sparse queries and applies a scalable transformer directly to these queries, allowing us to model geometry and context in a single, unified representation.
>
> ProtoOcc (AAAI 25) and ProtoOcc (CVPR 25) also group voxel features into scene-level or view-aligned prototypes. This has parallels to our approach, which groups individual Gaussians into sparse queries. However, our queries remain explicit in 3D, and their associated Gaussians can be repositioned or reconfigured both individually and as a group, enabling flexible, fine-grained modeling that is not directly possible with grid-aligned voxel prototypes.
>
> Finally, StreamOcc proposes an efficient 3D voxel propagation model coupled with explicit foreground instance queries. While a strong method, passing features between dense 3D voxels and instance-level box regions incurs a significant runtime cost. S2GO instead represents both foreground and background with sparse queries, providing a single, unified representation for the entire scene and avoiding separate voxel/query pathways.
>
> ## [W2]
> > Robustness of Geometric Prior: The pretraining's reliance on high-quality geometric data (LiDAR or strong depth models like Metric3D) is a concern. The analysis would be more robust with sensitivity studies on (a) varying LiDAR sparsity (e.g., 16 vs. 64-beam) and (b) the accuracy of the depth estimation input.
>
> First, we want to emphasize that our use of LiDAR during pretraining does not constitute an unfair advantage. LiDAR is used to generate the occupancy labels that all methods used for training, and many works (GaussianFormer-2, FB-OCC, GSD-Occ) also use LiDAR during the training process. Second, we note that Metric3D is a representative, but dated, depth estimation method, and we chose it to demonstrate the robustness of our framework. Recent methods, such as MoGe-2 (Wang et al.) demonstrate substantially better depth estimates.
>
> However, we add two additional results to Table 8 to demonstrate that our method can adapt well to other settings. We have copied Table 8 here:
>
> **Table 8. Ablation on pretraining with various depth sources.**
>
> | Pretraining Query Initialization        | mIoU  | IoU   |
> |----------------------------------------|------:|------:|
> | Occupied voxel locations               | 21.61 | 33.75 |
> | LiDAR (32-line)                        | 21.60 | 33.91 |
> | LiDAR (16-line)                        | 21.16 | 33.39 |
> | Zero-shot RGB depth pred. (Yin et al., 2023) | 20.99 | 33.57 |
>
> First, to account for the possibility where raw LiDAR is unavailable for model training (e.g. potentially occupancy was generated with a monocular or an onboard pipeline), we investigate using the occupied voxel locations directly instead of LiDAR. We observe no significant change in the results, demonstrating our framework is sufficiently robust. Second, if occupancy is not available during pretraining, we also investigate pretraining with a sparser 16-line LiDAR instead of nuScenes' default 32-line sensor. We simulate this by dropping out every other scan line. We find that we largely maintain performance even with such a cheaper sensor, with a 0.44 drop in mIoU. These results demonstrate that our framework can handle a variety of different possible pre-training settings.

---

### Official Review · Reviewer_GjMC · 2025-11-02

**Soundness:** 3
**Presentation:** 4
**Contribution:** 3
**Rating:** 6
**Confidence:** 5

**Summary:**

This paper presents a new method for sparse occupancy prediction. It proposes several methodological and engineering improvements over prior Gaussian-based approaches, including a refined pretraining strategy and an innovative Gaussian-to-Voxel Splatting operator. These contributions demonstrate both conceptual insight and practical significance. The proposed method achieves superior performance compared to previous Gaussian-based methods.

**Strengths:**

1. The paper is well-organized and written with clear logical flow.

2. It conducts an in-depth exploration of sparse occupancy prediction, proposing improvements such as pretraining strategies and the Gaussian-to-Voxel Splatting operator that are both insightful and practically valuable.

3. The method achieves better performance than prior Gaussian-based approaches, demonstrating tangible benefits of the proposed design.

**Weaknesses:**

1. Inference speed comparison (Tab. 1) may be unfair: the proposed method uses 256×704 input images, while other methods use 900×1600. Since the image backbone usually dominates inference time, comparison should be made at the same resolution or explicitly clarified in the paper/table.

2. Methods such as [2] achieve 25.5 mIoU on the SurroundOcc dataset but are not included in the comparison table.

3. While reporting results on Occ3D is commendable, the paper omits several recent competitive baselines with higher RayIoU (e.g., [1], [3], [4]).

4. Minor: L242 — “where M is the # of LiDAR points” appears to contain a small typo.

[1] OPUS: Occupancy Prediction Using a Sparse Set, NeurIPS 2024

[2] Rethinking Temporal Fusion with a Unified Gradient Descent View for 3D Semantic Occupancy Prediction, CVPR 2025

[3] STCOcc: Sparse Spatial-Temporal Cascade Renovation for 3D Occupancy and Scene Flow Prediction, CVPR 2025

[4] ALOcc: Adaptive Lifting-based 3D Semantic Occupancy and Cost Volume-based Flow Prediction, ICCV 2025

**Questions:**

1. The method achieves excellent results using lower-resolution inputs. Could the authors report performance using 900×1600 image size to compare under identical conditions with previous Gaussian-based methods?

2. How is the velocity of Gaussians predicted? Is there explicit velocity supervision during training?

3. In the ablation tables (Tab. 3–5), the best-performing configurations achieve different absolute scores. Could the authors clarify the cause of these variations?

4. Could sparse-based methods potentially outperform BEV-encoding methods? For example, ALOcc [4] achieves 39.3 RayIoU and 30.5 FPS on Occ3D-nuScenes, appearing more efficient.

---

> ### Author Response · Authors · 2025-11-24
>
> We sincerely thank the reviewer for their feedback, and we appreciate that they found our pretraining strategy and refinement of the splatting operation insightful and valuable. We have made updates to the manuscript in blue, and below, we address the reviewer's concerns in detail.
>
> ## [W1] & [Q1]
> > Inference speed and performance comparison using the same image resolution.
>
> Due to computational limitations, we were unable to train a high-resolution (900 x 1600) version of the model in time. However, following the reviewer's recommendation, we present a same-resolution comparison between our model and QuadricFormer, a very recent primitive-efficient method, in Appendix H. We update QuadricFormer to use the same ResNet-50 backbone as S2GO with an image size of 256x704.
>
> **Table 9. Quantitative comparisons with QuadricFormer (256×704). **
>
> | Method | IoU | mIoU | barrier | bicycle | bus | car | const. | motor. | ped. | traff. cone. | trailer | truck | drive. suf. | other flat | sidewalk | terrain | manmade | vegetation | FPS |
> |--------|----:|-----:|--------:|--------:|----:|----:|------------:|-----------:|-----------:|--------------:|--------:|------:|-------------:|-----------:|----------:|--------:|---------:|-----------:|----:|
> | **QuadricFormer-3.2k** | 28.9 | 18.5 | 18.0 | 11.0 | 25.7 | 28.0 | 11.0 | 13.3 | 11.7 | 9.1 | 11.5 | 19.2 | 39.0 | 23.1 | 24.5 | 22.9 | 9.4 | 19.2 | 22.3 |
> | **QuadricFormer-12.8k** | 30.3 | 18.9 | 18.2 | 10.6 | 25.3 | 28.4 | 11.0 | 12.6 | 11.6 | 9.4 | 12.3 | 19.8 | 39.9 | 22.6 | 25.6 | 23.7 | 10.8 | 19.8 | 13.9 |
> | **S2GO-Small** | 34.3 | 22.1 | 20.8 | 13.1 | 27.5 | 30.3 | 14.5 | 16.5 | 11.7 | 10.9 | 13.5 | 23.3 | 46.3 | 29.2 | 29.7 | 28.4 | 13.0 | 25.1 | **32.2** |
> | **S2GO-Base** | **35.5** | **22.7** | 21.9 | 13.4 | 27.5 | 32.1 | 14.9 | 15.3 | 12.9 | 11.8 | 13.4 | 24.0 | 46.9 | 29.1 | 30.3 | 29.1 | 14.7 | 26.4 | 23.6 |
>
>
> S2GO maintains a significantly better Pareto frontier, with S2GO-Base being slightly faster than QuadricFormer-3.2k (23.6 vs 22.3 FPS) while improving IoU by $\textbf{+6.6}$ points and mIoU by $\textbf{+4.2}$ points. We note that S2GO-Small maintains a substantial lead in both FPS and mIoU ($\textbf{+9.9}$ FPS and $\textbf{+3.6}$ mIoU), demonstrating S2GO's efficiency and performance independent of image resolution. Furthermore, for clarity, we updated Section 4 with discussion on the image resolution as well.
>
> ## [W2]
> > Methods such as [2] achieve 25.5 mIoU on the SurroundOcc dataset but are not included in the comparison table.
>
> We thank the reviewer for pointing out this omission. We have now added [2] to the SurroundOcc comparison in Table 1. This method does attain higher mIoU (25.5) on SurroundOcc, but at the cost of substantially lower throughput (0.9 FPS). In contrast, our proposed S2GO operates in a real-time setting using a sparse Gaussian representation.
>
> We have revised the text in Section 4.1 to clarify that our focus is on efficient, sparse Gaussian-based occupancy estimation. On SurroundOcc and SSCBench-KITTI-360, S2GO achieves state-of-the-art performance within this efficient regime while maintaining real-time throughput, whereas [2] targets a different performance–efficiency regime with much higher cost. We now explicitly discuss this trade-off when presenting Table 1.
>
> ## [W3]
> > While reporting results on Occ3D is commendable, the paper omits several recent competitive baselines with higher RayIoU (e.g., [1], [3], [4]).
>
> We thank the reviewer for these references; we have now included [1], [3], [4] and other methods in the updated Table 10 of the supplementary and report their RayIoU on Occ3D-nuScenes. These methods indeed achieve higher RayIoU than ours on this benchmark. In this work, however, our primary focus is on the SurroundOcc-nuScenes and SSCBench-KITTI-360 benchmarks, which we adopt as the main datasets for studying sparse Gaussian streaming occupancy. On these benchmarks, S2GO consistently improves over prior Gaussian-based models in both accuracy and throughput, indicating that our sparse Gaussian representation, pretraining strategy, and Gaussian-to-voxel splatting lead to a stronger accuracy–efficiency trade-off in the streaming setting. We now state this scope more explicitly in Section 4.
>
> We have also included a comparison of the ground truth annotations between SurroundOcc and Occ3D in Figure 10 of the supplementary. While both datasets are strong testbeds for semantic occupancy, SurroundOcc covers a larger spatial extent (100m $\times$ 100m vs.\ 80m $\times$ 80m), often offers more complete ground truth, and it serves as the main benchmark for the baselines that our method builds upon. We present Occ3D as a secondary benchmark evaluation. S2GO is not tuned specifically for Occ3D, yet still attains competitive RayIoU while maintaining high throughput.
>
> ## [W4]
> > Minor: L242 — “where M is the # of LiDAR points” appears to contain a small typo.
>
> Thank you for finding this; we have corrected this in the manuscript.

---

> ### Author Response · Authors · 2025-11-24
>
> ## [Q2]
> > How is the velocity of Gaussians predicted? Is there explicit velocity supervision during training?
>
> Each query predicts a velocity which is passed on to its constituent Gaussians. The velocity is learned in a self-supervised fashion. During both pretraining and occupancy training, Gaussians are supervised in neighboring frames as well (e.g. depth rendering or occupancy splatting), after being offset by its predicted velocity. We find that this is sufficient to learn reasonable motion of individual objects as shown in Figure 4.
>
> ## [Q3]
> > In the ablation tables (Tab. 3–5), the best-performing configurations achieve different absolute scores. Could the authors clarify the cause of these variations?
>
> The ablations in Tables 4 and 5 were performed prior to adding the denoising objective during pretraining. Further, the ablation in Table 4 was performed with a fixed $\delta$ during training and inference, without the random 0m to 3m sampling augmentation during training (described in Section B), which was done later for Tables 3 and 5.
>
> ## [Q4]
> > Could sparse-based methods potentially outperform BEV-encoding methods? For example, ALOcc [4] achieves 39.3 RayIoU and 30.5 FPS on Occ3D-nuScenes, appearing more efficient.
>
> We agree that BEV-encoding methods like ALOcc [4] are very strong on Occ3D-nuScenes, achieving both high RayIoU and high throughput. In this work, we investigate whether a sparse query–based streaming representation can achieve competitive performance–efficiency on a dense task such as semantic occupancy estimation over time.
>
> In this setting, sparse query representations offer different advantages compared to BEV or grid encoding methods. First, each S2GO query directly controls a group of Gaussians and carries an explicit velocity, which makes it natural to roll out future states by propagating individual elements forward in time. Second, computation is concentrated on occupied regions, so the network processes only a compact set of active queries instead of updating a dense BEV grid at every step, which can help it scale efficiently as resolution increases. Our experiments on SurroundOcc and SSCBench-KITTI-360 indicate that with our improvements, sparse query–based designs can be competitive with grid-based methods for semantic occupancy estimation, while providing a different and more flexible mechanism for temporal modeling.

---

> > ### Comment · Reviewer_GjMC · 2025-11-25
> >
> > I thank the authors for their detailed response and revisions. I recognize the value of this work and tend to increase my rating. However, I have two remaining concerns:
> > 1. Although I understand that running new experiments at 900x1600 is time consuming, it is critical to explicitly annotate the input resolutions for all methods in the tables or captions. Input size significantly dictates inference latency, as large inputs cause the backbone’s processing time to significantly overshadow the cost of the occupancy prediction modules (the authors could evaluate FPS with the input size increased to 900x1600 while keeping all other conditions constant.). Since efficiency is a core claim of this paper, comparing your method (at a lower resolution) against baselines (e.g., at 900x1600 in Tab. 1) without clear notation is misleading to readers.
> > 2. The use of different GPUs (RTX 4090, A6000, and A100) for FPS evaluation across Tabs. 1, 6, and 10 is inconsistent. I strongly suggest standardizing the hardware for all evaluations. This ensures a valid comparison across methods under different datasets (e.g., SurroundOcc and Occ3D).

---

> ### Author Response · Authors · 2025-11-27
>
> We thank the reviewer for their positive comments and feedback. We understand that precise reporting of latency is critical, and we have made a number of updates to the manuscript.
>
> > Although I understand that running new experiments at 900x1600 is time consuming, it is critical to explicitly annotate the input resolutions for all methods in the tables or captions. Input size significantly dictates inference latency, as large inputs cause the backbone’s processing time to significantly overshadow the cost of the occupancy prediction modules (the authors could evaluate FPS with the input size increased to 900x1600 while keeping all other conditions constant.). Since efficiency is a core claim of this paper, comparing your method (at a lower resolution) against baselines (e.g., at 900x1600 in Tab. 1) without clear notation is misleading to readers.
>
> We appreciate this valuable feedback and have further clarified the resolution settings in the revision. We now also explicitly emphasize in the caption of Table 1 that baselines use 900 $\times$ 1600 input resolution while S2GO uses 256 $\times$ 704. This information is also summarized in Section 4.1 when the results are discussed, which also directs readers to Appendix H for a same-resolution comparison with QuadricFormer.
>
> Following the reviewer’s suggestion, we additionally benchmark the inference speed of a high-resolution S2GO-Base version. When evaluated at the same 900x1600 resolution as QuadricFormer (6.2 FPS), S2GO-Base runs at 130.2 ms (7.7 FPS) on a single RTX 4090. As the reviewer mentioned, in this high-resolution setting, the backbone computation dominates overall latency and the relative cost of the occupancy head is smaller. We emphasize two points regarding this. First, at 900$\times$1600 the backbone accounts for most of the runtime, so differences between occupancy heads have a smaller impact on overall FPS. However, at the lower-resolution setting where the head is a larger fraction of the cost, our sparse query design is precisely what allows S2GO to run in real time while achieving strong accuracy. Second, even with lower resolution input, S2GO substantially outperforms prior high-resolution Gaussian-based methods and occupies a much stronger position on the overall performance–efficiency Pareto curve. This information is also included in Appendix H of the manuscript.
>
> > The use of different GPUs (RTX 4090, A6000, and A100) for FPS evaluation across Tabs. 1, 6, and 10 is inconsistent. I strongly suggest standardizing the hardware for all evaluations. This ensures a valid comparison across methods under different datasets (e.g., SurroundOcc and Occ3D).
>
> We sincerely apologize for the confusion caused by the use of different GPU types. In this revision, we have tried to make the hardware setup across experiments more consistent and easier to interpret:
>
> 1) We leave Table 1 as-is using a 4090 GPU because this is the standard GPU used for FPS reporting in prior work on SurroundOcc.
> 2) We update Table 6 (which previously used an A100 for training and a 6000 for inference) so that both training and inference are benchmarked on a 4090 GPU to be consistent with Table 1. The trends and takeaways remain unchanged.
> 3) For Table 10, different baselines report FPS on different GPUs, so we now include S2GO’s FPS on both an A100 and a 4090 GPU to enable a more complete comparison.

---

> > ### Comment · Reviewer_GjMC · 2025-11-27
> >
> > My concerns have been solved, so I raised my rating.

---

### Meta-Review · Area_Chair_iMMF · 2025-12-05

**Summary:**

The reviewers broadly agreed that the paper presents a novel and effective streaming sparse Gaussian framework with strong empirical gains and meaningful engineering contributions. The primary issues raised concerned fairness of efficiency comparisons, completeness of baselines, reliance on LiDAR-based pretraining, semantic and temporal behavior of queries, and memory/latency reporting. After rebuttal, most concerns were resolved to the satisfaction of reviewers, including the most critical ones (input resolution, hardware consistency). The remaining concerns are minor or conceptual and do not fundamentally challenge the validity of the work. Overall reviewer sentiment after rebuttal supports acceptance.

**Reviewer Concerns:**

**Addressed**

1. Fairness of efficiency comparisons: Authors explicitly annotated input resolutions, added a same-resolution comparison, and benchmarked S2GO at 900×1600.
2. Hardware consistency in FPS reporting: GPU usage was standardized; FPS for multiple GPUs reported for clarity.
3. Missing baselines (voxel & recent occupancy methods): Additional baselines added and trade-offs discussed (Table 10).
4. Pretraining robustness: Added experiments with 16-line LiDAR, occupancy-only pretraining, and monocular-depth pretraining (Table 8).
5. Opacity-weighted splatting cost: Provided complete memory and speed breakdown (Table 6) and 3.3× memory reduction via CUDA kernels.
6. Query identity/drift analysis: Added quantitative lifetime statistics (mean 9–15 frames).
7. Sensitivity to δ and queue length: Added plots showing mIoU vs. hyperparameters.
8. Visualization/layout and module clarity: Figures revised; architecture details expanded.

Outstanding (Minor)

1. Limited theoretical discussion: Representation power of sparse queries and convergence properties remain only lightly discussed; reviewers treated this as non-blocking.
2. Semantic rigidity within queries: Authors clarified behavior via opacity/scale mechanisms; deeper exploration or per-Gaussian semantics remains future work.
3. Domain shift generalization: Cross-dataset pretraining results added, but broader analysis is still limited.

**Reviewer Scores:**

**Reviewer GjMC (Initial rating: 6; Likely final score: 8)**

Post-discussion stance: The reviewer explicitly states that they raised their rating.

**Reviewer Hic9 (Initial Rating: 8; Likely final score: 8)**

The reviewer remained satisfied.

**Reviewer RwQB (Initial Rating: 6; Likely final score: 6)**

Minor concerns remain but the rebuttal addressed major issues; tone suggests they remain at a marginal-accept level.

**Reviewer m6uz (Initial Rating: 8; Likely final score: 8)**

The reviewer was already highly supportive and concerns were addressed.

---

### Decision · Program_Chairs · 2026-01-26

Accept (Poster)